# IMPROVED SAMPLING VIA LEARNED DIFFUSIONS

**Lorenz Richter**[*]
Zuse Institute Berlin
dida Datenschmiede GmbH
`richter@zib.de`

**Julius Berner**[*]
Caltech
`jberner@caltech.edu`

## ABSTRACT

Recently, a series of papers proposed deep learning-based approaches to sample from target distributions using controlled diffusion processes, being trained only on the unnormalized target densities without access to samples. Building on previous work, we identify these approaches as special cases of a generalized Schrödinger bridge problem, seeking a stochastic evolution between a given prior distribution and the specified target. We further generalize this framework by introducing a variational formulation based on divergences between path space measures of time-reversed diffusion processes. This abstract perspective leads to practical losses that can be optimized by gradient-based algorithms and includes previous objectives as special cases. At the same time, it allows us to consider divergences other than the reverse Kullback-Leibler divergence that is known to suffer from mode collapse. In particular, we propose the so-called *log-variance loss*, which exhibits favorable numerical properties and leads to significantly improved performance across all considered approaches.

## 1 INTRODUCTION

Given a function $\rho \colon \mathbb{R}^d \to [0, \infty)$, we consider the task of sampling from the density

$$p_{\text{target}} := \frac{\rho}{Z} \quad \text{with} \quad Z := \int_{\mathbb{R}^d} \rho(x)\,\mathrm{d}x,$$

where the normalizing constant $Z$ is typically intractable. This represents a crucial and challenging problem in various scientific fields, such as Bayesian statistics, computational physics, chemistry, or biology, see, e.g., Liu & Liu (2001); Stoltz et al. (2010). Fueled by the success of *denoising diffusion probabilistic modeling* (Song & Ermon, 2020; Ho et al., 2020; Kingma et al., 2021; Vahdat et al., 2021; Nichol & Dhariwal, 2021) and deep learning approaches to the *Schrödinger bridge* (SB) problem (De Bortoli et al., 2021; Chen et al., 2021a; Koshizuka & Sato, 2022), there is a significant interest in tackling the sampling problem by using stochastic differential equations (SDEs) which are controlled with learned neural networks to transport a given prior density $p_{\text{prior}}$ to the target $p_{\text{target}}$.

Recent works include the *Path Integral Sampler* (PIS) and variations thereof (Tzen & Raginsky, 2019; Richter, 2021; Zhang & Chen, 2022; Vargas et al., 2023b), the *Time-Reversed Diffusion Sampler* (DIS) (Berner et al., 2024), as well as the *Denoising Diffusion Sampler* (DDS) (Vargas et al., 2023a). While the ideas for such sampling approaches based on controlled diffusion processes date back to earlier work, see, e.g., Dai Pra (1991); Pavon (1989), the development of corresponding numerical methods based on deep learning has become popular in the last few years.

However, up to now, more focus has been put on *generative modeling*, where samples from $p_{\text{target}}$ are available. As a consequence, it seems that for the classical sampling problem, i.e., having only an analytical expression for $\rho \propto p_{\text{target}}$, but no samples, diffusion-based methods cannot reach state-of-the-art performance yet. Potential drawbacks might be stability issues during training, the need to differentiate through SDE solvers, or mode collapse due to the usage of objectives based on reverse Kullback-Leibler (KL) divergences, see, e.g., Zhang & Chen (2022); Vargas et al. (2023a).

In this work, we overcome these issues and advance the potential of sampling via learned diffusion processes toward more challenging problems. Our contributions can be summarized as follows:

---

[*]Equal contribution (the author order was determined by `numpy.random.rand(1)`).

- We provide a unifying framework for sampling based on learned diffusions from the perspective of measures on path space and time-reversals of controlled diffusions, which for the first time connects methods such as SB, DIS, DDS, and PIS.

- This path space perspective, in consequence, allows us to consider arbitrary divergences for the optimization objective, whereas existing methods solely rely on minimizing a reverse KL divergence, which is prone to mode collapse.

- In particular, we propose the log-variance divergence that avoids differentiation through the SDE solver and allows to balance exploration and exploitation, resulting in significantly improved numerical stability and performance, see Figure 1.

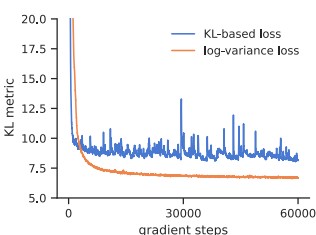

Figure 1: Improved convergence of our proposed log-variance loss for a double well problem, see Section 4 for further details.

## 1.1 RELATED WORK

There exists a myriad of Monte Carlo-based methods for sampling from unnormalized densities, e.g. *Annealed Importance Sampling* (AIS) (Neal, 2001), *Sequential Monte Carlo* (Del Moral et al., 2006; Doucet et al., 2009) (SMC), or *Markov chain Monte Carlo* (MCMC) (Kass et al., 1998). Note, however, that MCMC methods are usually only guaranteed to reach the target density asymptotically, and the convergence speed might be too slow in many practical settings (Robert et al., 1999). Variational methods such as *mean-field approximations* (Wainwright et al., 2008) and *normalizing flows* (Papamakarios et al., 2021; Wu et al., 2020; Midgley et al., 2023; Vaitl et al., 2024) provide an alternative. Similar to our setting, the problem of density estimation is cast into an optimization problem by fitting a parametric family of tractable distributions to the target density.

We build our theoretical foundation on the variational formulation of bridge problems proposed by Chen et al. (2021a). We recall that the underlying ideas were established decades ago (Nelson, 1967; Anderson, 1982; Haussmann & Pardoux, 1986; Föllmer, 1988), however, only recently applied to diffusion models (Song et al., 2020) and SBs (Vargas, 2021; Liu et al., 2022). While the numerical treatment of SB problems has classically been approached via iterative nested schemes, the approach in Chen et al. (2021a) uses backward SDEs (BSDEs) to arrive at a single objective based on a KL divergence. This objective includes the (continuous-time) ELBO of diffusion models (Huang et al., 2021) as a special case, which can also be approached from the perspective of optimal control (Berner et al., 2024). For additional previous work on optimal control in the context of generative modeling, we refer to De Bortoli et al. (2021); Tzen & Raginsky (2019); Pavon (2022); Holdijk et al. (2022).

Crucially, we note that our path space perspective on the variational formulation of bridges has not been known before. Our novel derivation only relies on time-reversals of diffusion processes and shows that, in general, corresponding losses (in particular the one in Chen et al. (2021a)) do not have a unique solution as they lack the entropy constraint of classical SB problems. However, in special cases, we recover unique objectives corresponding to recently developed sampling methods, e.g., DIS, DDS, and PIS. Moreover, the path space perspective allows us to extend the variational formulation of bridges to different divergences, in particular to the log-variance divergence that has originally been introduced in Nüsken & Richter (2021). Variants of this loss have previously only been analyzed in the context of variational inference (Richter et al., 2020) and neural solvers for partial differential equations (PDEs) (Richter & Berner, 2022). Extending these works, we prove that the beneficial properties of the log-variance loss also hold for the general bridge objective, which incorporates two instead of only one controlled stochastic process. Finally, we refer to Vargas et al. (2024) for concurrent work on the path space perspective on diffusion-based sampling.

## 1.2 OUTLINE OF THE ARTICLE

The rest of the article is organized as follows. In Section 2, we provide an introduction to diffusion-based sampling from the perspective of path space measures and time-reversed SDEs. This can be understood as a unifying framework allowing for generalizations to divergences other than the KL divergence. We propose the *log-variance divergence* and prove that it exhibits superior properties. In Section 3, we will subsequently outline multiple novel connections to known methods, such as SBs in Section 3.1, diffusion-based generative modeling (i.e., DIS) in Section 3.2, and approaches based on reference processes (i.e., PIS and DDS) in Section 3.3. For all considered methods, we can find compelling numerical evidence for the superiority of the log-variance divergence, see Section 4.

## 2 DIFFUSION-BASED SAMPLING

In this section, we will reformulate our sampling problem as a time-reversal of diffusion processes from the perspective of measures on path space. Let us first define our setting and notation.

### 2.1 NOTATION AND SETTING

We denote the density of a random variable $X$ by $p_X$. For a suitable $\mathbb{R}^d$-valued stochastic process $X = (X_t)_{t \in [0,T]}$ we define its density $p_X$ w.r.t. to the Lebesgue measure by $p_X(\cdot, t) := p_{X_t}$ for $t \in [0, T]$. For suitable functions $f \in C(\mathbb{R}^d \times [0, T], \mathbb{R})$ and $w \in C(\mathbb{R}^d \times [0, T], \mathbb{R}^d)$, we further define the deterministic and stochastic integrals

$$R_f(X) := \int_0^T f(X_s, s)\, \mathrm{d}s \qquad \text{and} \qquad S_w(X) := \int_0^T w(X_s, s) \cdot \mathrm{d}W_s, \tag{1}$$

where $W$ is a standard $d$-dimensional Brownian motion. We denote by $\mathcal{P}$ the set of all probability measures on the space of continuous functions $C([0, T], \mathbb{R}^d)$ and define the path space measure $\mathbb{P}_X \in \mathcal{P}$ as the law of $X$. For a time-dependent function $\mu$, we denote by $\breve{\mu}$ the time-reversal given by $\breve{\mu}(t) := \mu(T - t)$. We refer to Appendix A.1 for technical assumptions.

### 2.2 SAMPLING AS TIME-REVERSAL PROBLEM

The goal of diffusion-based sampling is to sample from the density $p_{\text{target}} = \frac{\rho}{Z}$ by transporting a prior density $p_{\text{prior}}$ via controlled stochastic processes. We consider two processes given by the SDEs

$$\mathrm{d}X_s^u = (\mu + \sigma u)(X_s^u, s)\, \mathrm{d}s + \sigma(s)\, \mathrm{d}W_s, \qquad\qquad X_0^u \sim p_{\text{prior}}, \tag{2}$$
$$\mathrm{d}Y_s^v = (-\breve{\mu} + \breve{\sigma}\breve{v})(Y_s^v, s)\, \mathrm{d}s + \breve{\sigma}(s)\, \mathrm{d}W_s, \qquad\qquad Y_0^v \sim p_{\text{target}}, \tag{3}$$

where we aim to identify control functions $u, v \in \mathcal{U}$ in a suitable space of admissible controls $\mathcal{U} \subset C(\mathbb{R}^d \times [0, T], \mathbb{R}^d)$ in order to achieve $X_T^u \sim p_{\text{target}}$ and $Y_T^v \sim p_{\text{prior}}$. Specifically, we seek controls satisfying

$$p_{\text{prior}} \overset{X^u}{\underset{Y^v}{\rightleftarrows}} p_{\text{target}}$$

in the sense that $Y^v$ is the time-reversed process of $X^u$ and vice versa, i.e., $\breve{p}_{X^u} = p_{Y^v}$. In this context, we recall the following well-known result on the time-reversal of stochastic processes (Nelson, 1967; Anderson, 1982; Haussmann & Pardoux, 1986; Föllmer, 1988).

**Lemma 2.1** (Time-reversed SDEs). *The time-reversed process $\overleftarrow{Y}^v$, given by the SDE*

$$\mathrm{d}\overleftarrow{Y}_s^v = \left(\mu - \sigma v + \sigma\sigma^\top \nabla \log \breve{p}_{Y^v}\right)(\overleftarrow{Y}_s^v, s)\, \mathrm{d}s + \sigma(s)\, \mathrm{d}W_s, \qquad \overleftarrow{Y}_0^v \sim Y_T^v, \tag{4}$$

*satisfies that $p_{\overleftarrow{Y}^v} = \breve{p}_{Y^v}$.*

*Proof.* The result can be derived by comparing the Fokker-Planck equations governing $p_{\overleftarrow{Y}^v}$ and $p_{Y^v}$, see, e.g., Chen et al. (2021a); Huang et al. (2021); Berner et al. (2024). $\square$

Let us now define the problem of identifying the desired control functions $u$ and $v$ from the perspective of path space measures on the space of trajectories $C([0, T], \mathbb{R}^d)$, as detailed in the sequel.

**Problem 2.2** (Time-reversal). Let $\mathbb{P}_{X^u}$ be the path space measure of the process $X^u$ defined in (2) and let $\mathbb{P}_{\overleftarrow{Y}^v}$ be the path space measure of $\overleftarrow{Y}^v$, the time-reversal of $Y^v$, given in (4). Further, let $D : \mathcal{P} \times \mathcal{P} \to \mathbb{R}_{\geq 0}$ be a divergence, i.e., a non-negative function satisfying that $D(\mathbb{P}, \mathbb{Q}) = 0$ if and only if $\mathbb{P} = \mathbb{Q}$. We aim to find optimal controls $u^*, v^*$ s.t.

$$u^*, v^* \in \underset{u,v \in \mathcal{U} \times \mathcal{U}}{\arg\min}\, D\left(\mathbb{P}_{X^u} | \mathbb{P}_{\overleftarrow{Y}^v}\right). \tag{5}$$

We note that Problem 2.2 aims to reverse the processes $X^u$ and $Y^v$ with respect to each other while obeying the respective initial values $X_0^u \sim p_{\text{prior}}$ and $Y_0^v \sim p_{\text{target}}$. For the actual computation of suitable divergences, we derive the following fundamental identity.

**Proposition 2.3** (Likelihood of path measures). *Let $X^w$ be a process as defined in (2) with $u$ being replaced by $w \in \mathcal{U}$ and let $S$ and $R$ be as in (1). We can compute the Radon-Nikodym derivative as*

$$\frac{\mathrm{d}\mathbb{P}_{X^u}}{\mathrm{d}\mathbb{P}_{\tilde{Y}^v}}(X^w) = Z \exp\left(R_{f_{u,v,w}^{\mathrm{Bridge}}} + S_{u+v} + B\right)(X^w) \tag{6}$$

*with* $\qquad B(X^w) := \log \dfrac{p_{\mathrm{prior}}(X_0^w)}{\rho(X_T^w)} \qquad$ *and* $\qquad f_{u,v,w}^{\mathrm{Bridge}} := (u+v) \cdot \left(w + \dfrac{v-u}{2}\right) + \nabla \cdot (\sigma v - \mu).$

*Proof.* The proof combines Girsanov's theorem, Itô's lemma, the HJB equation governing $\log p_{\tilde{Y}^v}$, and the fact that $\rho = Z p_{\mathrm{target}}$, see Appendix A.2. $\qquad\square$

Note that we can remove the divergence $\nabla \cdot (\sigma v - \mu)$ in (6) by resorting to backward stochastic integrals, see Remark A.1 in the appendix. Using the path space perspective and the representation of the Radon-Nikodym derivative in Proposition 2.3, we may now, in principle, choose any suitable divergence $D$ in order to approach Problem 2.2. Using our path space formulation, we are, to the best of our knowledge, the first to study this problem in such generality. In the following, we demonstrate that this general framework unifies previous approaches and allows us to derive new methods easily.

## 2.3 COMPARISON OF THE KL AND LOG-VARIANCE DIVERGENCE

Most works in the context of diffusion-based sampling rely on the KL divergence. Choosing $D = D_{\mathrm{KL}}$, which implies $w = u$ in (6), we can readily compute

$$D_{\mathrm{KL}}(\mathbb{P}_{X^u}|\mathbb{P}_{\tilde{Y}^v}) = \mathbb{E}\left[\left(R_{f_{u,v,u}^{\mathrm{Bridge}}} + B\right)(X^u)\right] + \log Z$$

with $f_{u,v,u}^{\mathrm{Bridge}} = \frac{\|u+v\|^2}{2} + \nabla \cdot (\sigma v - \mu)$, where we used the fact that the stochastic integral $S_{u+v}$ has vanishing expectation. Note that in practice we minimize the objective

$$\mathcal{L}_{\mathrm{KL}}(u,v) := D_{\mathrm{KL}}(\mathbb{P}_{X^u}|\mathbb{P}_{\tilde{Y}^v}) - \log Z. \tag{7}$$

This objective is analogous to the one derived in Chen et al. (2021a) for the bridge problem, see also Section 3.1 and Appendix A.4. Unfortunately, however, the KL divergence is known to have some evident drawbacks, such as mode collapse (Minka et al., 2005; Midgley et al., 2023) or a potentially high variance of Monte Carlo estimators (Roeder et al., 2017). To address those issues, we propose another divergence that has been originally suggested in Nüsken & Richter (2021) and extend it to the setting of two controlled stochastic processes.

**Definition 2.4** (Log-variance divergence). Let $\widetilde{\mathbb{P}}$ be a reference measure. The log-variance divergence between the measures $\mathbb{P}$ and $\mathbb{Q}$ w.r.t. the reference $\widetilde{\mathbb{P}}$ is defined as

$$D_{\mathrm{LV}}^{\widetilde{\mathbb{P}}}(\mathbb{P}, \mathbb{Q}) := \mathbb{V}_{\widetilde{\mathbb{P}}}\left[\log \frac{\mathrm{d}\mathbb{P}}{\mathrm{d}\mathbb{Q}}\right].$$

Note that the log-variance divergence is symmetric in $\mathbb{P}$ and $\mathbb{Q}$ and actually corresponds to a family of divergences, parametrized by the reference measure $\widetilde{\mathbb{P}}$. Obvious choices in our setting are $\widetilde{\mathbb{P}} := \mathbb{P}_{X^w}, \mathbb{P} := \mathbb{P}_{X^u}$, and $\mathbb{Q} := \mathbb{P}_{\tilde{Y}^v}$, resulting in the log-variance loss

$$\mathcal{L}_{\mathrm{LV}}^w(u,v) := D_{\mathrm{LV}}^{\mathbb{P}_{X^w}}(\mathbb{P}_{X^u}, \mathbb{P}_{\tilde{Y}^v}) = \mathbb{V}\left[\left(R_{f_{u,v,w}^{\mathrm{Bridge}}} + S_{u+v} + B\right)(X^w)\right]. \tag{8}$$

Since the variance is shift-invariant, we can omit $\log Z$ in the above objective.

Compared to the KL-based loss (7), the log-variances loss (8) exhibits the following beneficial properties. First, by the choice of the reference measure $\mathbb{P}_{X^w}$, one can balance exploitation and exploration. To exploit the current control $u$, one can set $w = u$, but one can also choose another control or another initial condition $X_0^w$. We can leverage this to counteract mode collapse by optimizing the loss in (8) along (sub-)trajectories where $\mathbb{P}_{X^u}$ has low probability, see Appendix A.7. Next, note that the log-variance loss in (8) does not require the derivative of the process $X^w$ w.r.t. the control $w$ (which, for the case $w = u$, is implemented by detaching or stopping the gradient, see Appendix A.6). While we still need to simulate the process $X^w$, we can rely on any (black-box) SDE solver and do not need to track the computation of $X^w$ for automatic differentiation. This

implies that the log-variance loss does not require derivatives[1] of the unnormalized target density $\rho$, which is crucial for problems where the target is only available as a black box. In contrast, the KL-based loss in (7) demands to differentiate $X^u$ w.r.t. $u$, requiring to differentiate through the SDE solver and resulting in higher computational costs. Particularly interesting is the following property, sometimes referred to as *sticking-the-landing* (Roeder et al., 2017). It states that the gradients of the log-variance loss have zero variance at the optimal solution. This property does, in general, not hold for the KL-based loss, such that variants of gradient descent might oscillate around the optimum.

**Proposition 2.5** (Robustness at the solution). *Let $\widehat{\mathcal{L}}_{\mathrm{LV}}$ be the Monte Carlo estimator of the log-variance loss in (8) and let the controls $u = u_\theta$ and $v = v_\gamma$ be parametrized by $\theta$ and $\gamma$. The variances of the respective derivatives vanish at the optimal solution $(u^*, v^*) = (u_{\theta^*}, v_{\gamma^*})$, i.e.*

$$\mathbb{V}\left[\nabla_\theta \widehat{\mathcal{L}}_{\mathrm{LV}}^w (u_{\theta^*}, v_{\gamma^*})\right] = 0 \qquad \textit{and} \qquad \mathbb{V}\left[\nabla_\gamma \widehat{\mathcal{L}}_{\mathrm{LV}}^w (u_{\theta^*}, v_{\gamma^*})\right] = 0,$$

*for all $w \in \mathcal{U}$. For the estimator $\widehat{\mathcal{L}}_{\mathrm{KL}}$ of the KL-based loss (7) the variances do not vanish.*

*Proof.* The proof is based on a technical calculation and Proposition 2.3, see Appendix A.2. $\square$

For the case $w = u$, we can further interpret the log-variance loss as a control variate version of the KL-based loss, see Remark A.2 in the appendix. We can empirically observe the variance reduction for the loss and its gradient in Figure 5 in the appendix.

## 3 CONNECTIONS AND EQUIVALENCES OF DIFFUSION-BASED SAMPLING APPROACHES

In general, there are infinitely many solutions to Problem 2.2 and, in particular, to our objectives in (7) and (8). In fact, Girsanov's theorem shows that the objectives only enforce Nelson's identity (Nelson, 1967), i.e.,

$$u^* + v^* = \sigma^\top \nabla \log p_{X^{u^*}} = \sigma^\top \nabla \log \overleftarrow{p}_{Y^{v^*}}, \tag{9}$$

see also the proof of Proposition 2.3. In this section, we show how our setting generalizes existing diffusion-based sampling approaches, which in turn ensure unique solutions to Problem 2.2. Moreover, with our framework, we can readily derive the corresponding versions of the log-variance loss (8). We refer to Appendix A.3 for a high-level overview of previous diffusion-based sampling methods.

### 3.1 SCHRÖDINGER BRIDGE PROBLEM (SB)

Out of all solutions $u^*$ fulfilling (9), the *Schrödinger bridge problem* considers the solution $u^*$ that minimizes the KL divergence $D_{\mathrm{KL}}(\mathbb{P}_{X^{u^*}}|\mathbb{P}_{X^r})$ to a given reference process $X^r$, defined as in (2) with $u$ replaced by $r \in \mathcal{U}$, see Appendix A.4 for further details. Traditionally, $r = 0$, i.e., the uncontrolled process $X^0$ is chosen. Defining

$$f_{u,r,w}^{\mathrm{ref}} := (u - r) \cdot \left(w - \frac{u + r}{2}\right), \tag{10}$$

Girsanov's theorem shows that $\frac{\mathrm{d}\mathbb{P}_{X^u}}{\mathrm{d}\mathbb{P}_{X^r}}(X^w) = \exp\left(R_{f_{u,r,w}^{\mathrm{ref}}} + S_{u-r}\right)(X^w)$, which implies that

$$D_{\mathrm{KL}}(\mathbb{P}_{X^u}|\mathbb{P}_{X^r}) = \mathbb{E}\left[R_{f_{u,r,u}^{\mathrm{ref}}}(X^u)\right], \tag{11}$$

see, e.g., Nüsken & Richter (2021, Lemma A.1) and the proof of Proposition 2.3. The SB objective can thus be written as

$$\min_{u \in \mathcal{U}} \mathbb{E}\left[R_{f_{u,r,u}^{\mathrm{ref}}}(X^u)\Big| X_T^u \sim p_{\mathrm{target}}\right], \tag{12}$$

see De Bortoli et al. (2021); Caluya & Halder (2021); Pavon & Wakolbinger (1991); Benamou & Brenier (2000); Chen et al. (2021b); Bernton et al. (2019). We note that the above can also be interpreted as an entropy-regularized *optimal transport* problem (Léonard, 2014). The entropy constraint in (11) could now be combined with our objective in (5) by considering, for instance,

$$\min_{u,v \in \mathcal{U} \times \mathcal{U}} \left\{ \mathbb{E}\left[R_{f_{u,r,u}^{\mathrm{ref}}}(X^u)\right] + \lambda D\left(\mathbb{P}_{X^u}|\mathbb{P}_{\overleftarrow{Y}^v}\right)\right\},$$

---

[1]While, by default, the samplers presented in the following use $\nabla \log \rho$ in their parametrization of the control $u$, we present promising results for the derivative-free regime in Appendix A.7.

where $\lambda \in (0, \infty)$ is a sufficiently large Lagrange multiplier. In Appendix A.4 we show how the SB problem (12) can be reformulated as a system of coupled PDEs or BSDEs, which can alternatively be used to regularize Problem 2.2, see also Liu et al. (2022); Koshizuka & Sato (2022). Interestingly, the BSDE system recovers our KL-based objective in (7), as originally derived in Chen et al. (2021a).

Note that via Nelson's identity (9), an optimal solution $u^*$ to the SB problem uniquely defines an optimal control $v^*$ and vice versa. For special cases of SBs, we can calculate such $v^*$ or an approximation $\bar{v} \approx v^*$. Fixing $v = \bar{v}$ in (5) and only optimizing for $u$ appearing in the generative process (2) then allows us to attain unique solutions to (an approximation of) Problem 2.2. We note that the approximation $\bar{v} \approx v^*$ incurs an irreducible loss given by

$$\frac{\mathrm{d}\mathbb{P}_{X^{u^*}}}{\mathrm{d}\mathbb{P}_{\bar{Y}^{\bar{v}}}}(X^w) = \frac{\mathrm{d}\mathbb{P}_{\bar{Y}^{v^*}}}{\mathrm{d}\mathbb{P}_{\bar{Y}^{\bar{v}}}}(X^w), \tag{13}$$

thus requiring an informed choice of $\bar{v}$ and $p_{\mathrm{prior}}$, such that $Y^{\bar{v}} \approx Y^{v^*}$. We will consider two such settings in the following sections.

## 3.2 DIFFUSION-BASED GENERATIVE MODELING (DIS)

We may set $\bar{v} := 0$, which can be interpreted as a SB with $u^* = r = \sigma^\top \nabla \log \overleftarrow{p}_{Y^0}$ and $p_{\mathrm{prior}} = p_{Y^0_T}$, such that the entropy constraint (11) can be minimized to zero. Note, though, that this only leads to feasible sampling approaches if the functions $\mu$ and $\sigma$ in the SDEs are chosen such that the distribution of $p_{Y^0_T}$ is (approximately) known and such that we can easily sample from it. In practice, one chooses functions $\mu$ and $\sigma$ such that $p_{Y^0_T} \approx p_{\mathrm{prior}} := \mathcal{N}(0, \nu^2 \mathrm{I})$, see Appendix A.6. Related approaches are often called *diffusion-based generative modeling* or *denoising diffusion probabilistic modeling* since the (optimally controlled) generative process $X^{u^*}$ can be understood as the time-reversal of the process $Y^0$ that moves samples from the target density to Gaussian noise (Ho et al., 2020; Pavon, 1989; Huang et al., 2021; Song et al., 2020). Let us recall the notation from Proposition 2.3 and define $f^{\mathrm{DIS}}_{u,w} := f^{\mathrm{Bridge}}_{u,0,w} = u \cdot w - \frac{\|u\|^2}{2} - \nabla \cdot \mu$. Setting $v = 0$ in (7), we readily get the loss

$$\mathcal{L}_{\mathrm{KL}}(u) = \mathbb{E}\left[\left(R_{f^{\mathrm{DIS}}_{u,u}} + B\right)(X^u)\right],$$

which corresponds to the *Time-Reversed Diffusion Sampler* (DIS) derived in Berner et al. (2024). Analogously, our path space perspective and (8) yield the corresponding log-variance loss

$$\mathcal{L}^w_{\mathrm{LV}}(u) = \mathbb{V}\left[\left(R_{f^{\mathrm{DIS}}_{u,w}} + S_u + B\right)(X^w)\right]. \tag{14}$$

## 3.3 TIME-REVERSAL OF REFERENCE PROCESSES (PIS & DDS)

In general, we may also set $\bar{v} := \sigma^\top \nabla \log p_{X^r} - r$. Via Lemma 2.1 this implies that $\mathbb{P}_{X^r} = \mathbb{P}_{\bar{Y}^{\bar{v},\mathrm{ref}}}$, where $Y^{v,\mathrm{ref}}$ is the process $Y^v$ as in (3), however, with $p_{\mathrm{target}}$ replaced by $p_{\mathrm{ref}} := p_{X^r_T}$, i.e.,

$$\mathrm{d}Y^{v,\mathrm{ref}} = (-\bar{\mu} + \bar{\sigma}\bar{v})(Y^{v,\mathrm{ref}}, s)\,\mathrm{d}s + \bar{\sigma}(s)\,\mathrm{d}W_s, \qquad Y^{v,\mathrm{ref}} \sim p_{\mathrm{ref}}.$$

In other words, $Y^{\bar{v},\mathrm{ref}}$ is the time-reversal of the reference process $X^r$. Using (6) with $p_{\mathrm{ref}}$ instead of $p_{\mathrm{target}} = \frac{\rho}{Z}$, we thus obtain that

$$1 = \frac{\mathrm{d}\mathbb{P}_{X^r}}{\mathrm{d}\mathbb{P}_{\bar{Y}^{\bar{v},\mathrm{ref}}}}(X^w) = \frac{p_{\mathrm{prior}}(X^w_0)}{p_{\mathrm{ref}}(X^w_T)} \exp\left(R_{f^{\mathrm{Bridge}}_{r,\bar{v},w}} + S_{r+\bar{v}}\right)(X^w). \tag{15}$$

This identity leads to the following alternative representation of Proposition 2.3.

**Lemma 3.1** (Likelihood w.r.t. reference process). *Assuming $\mathbb{P}_{X^r} = \mathbb{P}_{\bar{Y}^{\bar{v},\mathrm{ref}}}$, it holds that*

$$\frac{\mathrm{d}\mathbb{P}_{X^u}}{\mathrm{d}\mathbb{P}_{\bar{Y}^{\bar{v}}}}(X^w) = Z \exp\left(R_{f^{\mathrm{ref}}_{u,r,w}} + S_{u-r} + B^{\mathrm{ref}}\right)(X^w),$$

*where $f^{\mathrm{ref}}_{u,r,w}$ is defined as in (10) and $B^{\mathrm{ref}}(X^w) := \log \frac{p_{\mathrm{ref}}}{\rho}(X^w_T)$.*

*Proof.* The result follows from dividing $\frac{\mathrm{d}\mathbb{P}_{X^u}}{\mathrm{d}\mathbb{P}_{\bar{Y}^{\bar{v}}}}(X^w)$ in (6) by $\frac{\mathrm{d}\mathbb{P}_{X^r}}{\mathrm{d}\mathbb{P}_{\bar{Y}^{\bar{v},\mathrm{ref}}}}(X^w)$ in (15). We also refer to Remark A.3 for an alternative derivation that does not rely on the concept of time-reversals. □

Table 1: Comparison of the objectives with $R_f, S_u, B, B^{\mathrm{ref}}, f_{u,v,w}^{\mathrm{Bridge}}, f_{u,r,w}^{\mathrm{ref}}$ as defined in the text.

|  | $\mathcal{L}_{\mathrm{KL}}$ | $\mathcal{L}_{\mathrm{LV}}^w$ (ours) | $p_{\mathrm{prior}}$ | $v$ | $r$ |
|---|---|---|---|---|---|
| **Bridge** | $\mathbb{E}\left[\left(R_{f_{u,v,u}^{\mathrm{Bridge}}} + B\right)(X^u)\right]$ | $\mathbb{V}\left[\left(R_{f_{u,v,w}^{\mathrm{Bridge}}} + S_{u+v} + B\right)(X^w)\right]$ | arbitrary | learned | – |
| **DIS** | $\mathbb{E}\left[\left(R_{f_{u,0,u}^{\mathrm{Bridge}}} + B\right)(X^u)\right]$ | $\mathbb{V}\left[\left(R_{f_{u,0,w}^{\mathrm{Bridge}}} + S_u + B\right)(X^w)\right]$ | $\approx p_{Y_T^0}$ | $0$ | – |
| **PIS** | $\mathbb{E}\left[\left(R_{f_{u,0,u}^{\mathrm{ref}}} + B^{\mathrm{ref}}\right)(X^u)\right]$ | $\mathbb{V}\left[\left(R_{f_{u,0,w}^{\mathrm{ref}}} + S_u + B^{\mathrm{ref}}\right)(X^w)\right]$ | $\delta_{x_0}$ | $\sigma^\top \nabla \log p_{X^0}$ | $0$ |
| **DDS** | $\mathbb{E}\left[\left(R_{f_{u,r,u}^{\mathrm{ref}}} + B^{\mathrm{ref}}\right)(X^u)\right]$ | $\mathbb{V}\left[\left(R_{f_{u,r,w}^{\mathrm{ref}}} + S_{u-r} + B^{\mathrm{ref}}\right)(X^w)\right]$ | $p_{Y_T^{0,\mathrm{ref}}}$ | $0$ | $\sigma^\top \nabla \log \breve{p}_{Y^{0,\mathrm{ref}}}$ |

Note that computing the Radon-Nikodym derivative in Lemma 3.1 requires to choose $r, p_{\mathrm{prior}}, \mu$, and $\sigma$ such that $p_{\mathrm{ref}} = p_{X_T^r}$ is tractable[2]. For suitable choices of $r$ (see below), one can, for instance, use the SDEs with tractable densities stated in Appendix A.5 with $p_{\mathrm{prior}} = \delta_{x_0}, p_{\mathrm{prior}} = \mathcal{N}(0, \nu^2 \mathrm{I})$, or a mixture of such distributions. Recalling (13) and the choice $\bar{v} := \sigma^\top \nabla \log p_{X^r} - r$, we also need to guarantee that $Y^{\bar{v}} \approx Y^{v^*}$. Let us outline two such cases in the following.

**PIS:** We first consider the case $r := 0$. Lemma 3.1 and taking $D = D_{\mathrm{KL}}$ in Problem 2.2 then yields

$$\mathcal{L}_{\mathrm{KL}}(u) = D_{\mathrm{KL}}(\mathbb{P}_{X^u} | \mathbb{P}_{\bar{Y}^{\bar{v}}}) - \log Z = \mathbb{E}\left[\left(R_{f_{u,0,u}^{\mathrm{ref}}} + B^{\mathrm{ref}}\right)(X^u)\right].$$

This objective has previously been considered by Tzen & Raginsky (2019); Dai Pra (1991) and corresponding numerical algorithms, referred to as *Path Integral Sampler* (PIS) in Zhang & Chen (2022), have been independently presented in Richter (2021); Zhang & Chen (2022); Vargas et al. (2023b). Choosing $D = D_{\mathrm{LV}}$, we get the corresponding log-variance loss

$$\mathcal{L}_{\mathrm{LV}}^w(u) = \mathbb{V}\left[\left(R_{f_{u,0,w}^{\mathrm{ref}}} + S_u + B^{\mathrm{ref}}\right)(X^w)\right],$$

which has already been stated by Richter (2021, Example 7.1). Typically, the objectives are used with $p_{\mathrm{prior}} := \delta_{x_0}$, since Doob's $h$-transform guarantees that $\bar{v} = v^*$, i.e., we can solve the SB exactly, see Rogers & Williams (2000) and also Appendix A.4.1. In this special case, the SB is often referred to as a *Schrödinger half-bridge*.

**DDS:** Next, we consider the choices $r := \sigma^\top \nabla \log \breve{p}_{Y^{0,\mathrm{ref}}}, \bar{v} := 0$, and $p_{\mathrm{prior}} := p_{Y_T^{0,\mathrm{ref}}}$, which in turn yields a special case of the setting from Section 3.2. Using Lemma 3.1, we obtain the objective

$$\mathcal{L}_{\mathrm{KL}}(u) = \mathbb{E}\left[\left(R_{f_{u,r,u}^{\mathrm{ref}}} + B^{\mathrm{ref}}\right)(X^u)\right].$$

This corresponds to the *Denoising Diffusion Sampler* (DDS) objective stated by Vargas et al. (2023a) when choosing $\mu$ and $\sigma$ such that $Y^0$ is a VP SDE, see Appendix A.5. Choosing the invariant distribution $p_{\mathrm{ref}} := \mathcal{N}(0, \nu^2 \mathrm{I})$ of the VP SDE, see (26) in the appendix, we have that $p_{X^r}(\cdot, t) = \breve{p}_{Y^{0,\mathrm{ref}}}(\cdot, t) = p_{\mathrm{ref}} = p_{\mathrm{prior}}$ for $t \in [0, T]$, and, in particular, $r(x, t) = -\frac{\sigma^\top x}{\nu^2}$. Finally, with our general framework, the corresponding log-variance loss can now readily be computed as

$$\mathcal{L}_{\mathrm{LV}}^w(u) = \mathbb{V}\left[\left(R_{f_{u,r,w}^{\mathrm{ref}}} + S_{u-r} + B^{\mathrm{ref}}\right)(X^w)\right].$$

We refer to Table 1 for a comparison of all our different objectives.

## 4 NUMERICAL EXPERIMENTS

In this section, we compare the KL-based loss with the log-variance loss on the three different approaches, i.e., the general bridge, PIS, and DIS, introduced in Sections 2.3, 3.2, and 3.3. As DDS can be seen as a special case of DIS (both with $\bar{v} = 0$, see also Berner et al., 2024, Appendix A.10.1), we do not consider it separately. We can demonstrate that the appealing properties of the log-variance

---

[2]In general, it suffices to be able to compute $p_{X_T^r}$ up to its normalizing constant.

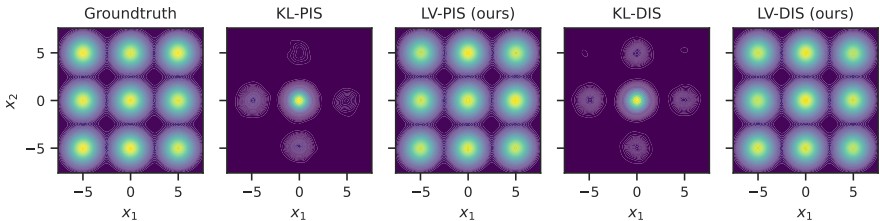

Figure 2: KDE plots of (1) samples from the groundtruth distribution, (2 & 3) PIS with KL divergence and log-variance loss, and (4 & 5) DIS with KL divergence and log-variance loss for the GMM problem (from left to right). One can see that the log-variance loss does not suffer from mode collapse such as the reverse KL divergence, which only recovers the mode of $p_{\text{prior}} = \mathcal{N}(0, \text{I})$.

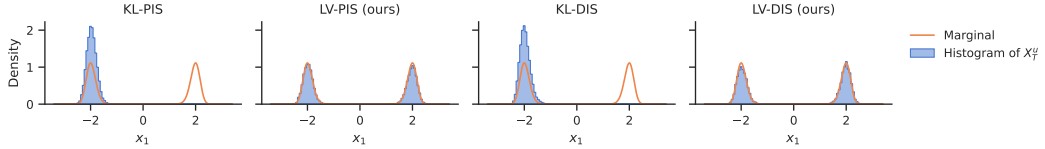

Figure 3: Marginals of the first coordinate of samples from PIS and DIS (left and right) for the DW problem with $d = 5$, $m = 5$, $\delta = 4$. Again, one observes the mode coverage of the log-variance loss as compared to the reverse KL divergence. Similar behavior can also be observed for the other marginals (see Figure 6) and higher-dimensional settings (see Figure 7 for an example in $d = 1000$).

loss can indeed lead to remarkable performance improvements for all considered approaches. Note that we always compare the same settings, in particular, the same number of target evaluations, for both the log-variance and KL-based losses and use sufficiently many gradient steps to reach convergence. See Appendix A.6 and Algorithm 1 for computational details[3]. Still, we observe that qualitative differences between the two losses are consistent across various hyperparameter settings. We refer to Appendix A.8 for additional experiments.

## 4.1 BENCHMARK PROBLEMS

We evaluate the different methods on the following three numerical benchmark examples.

**Gaussian mixture model (GMM):** We consider $\rho(x) = \frac{1}{m} \sum_{i=1}^{m} \mathcal{N}(x; \mu_i, \Sigma_i)$ and choose $m = 9$, $\Sigma_i = 0.3 \, \text{I}$, $(\mu_i)_{i=1}^{9} = \{-5, 0, 5\} \times \{-5, 0, 5\} \subset \mathbb{R}^2$ to obtain well-separated modes, see Figure 2.

**Funnel:** The 10-dimensional *Funnel distribution* (Neal, 2003) is a challenging example often used to test MCMC methods. It is given by the density $\rho(x) = p_{\text{target}}(x) = \mathcal{N}(x_1; 0, \eta^2) \prod_{i=2}^{d} \mathcal{N}(x_i; 0, e^{x_1})$ for $x = (x_i)_{i=1}^{10} \in \mathbb{R}^{10}$ with $\eta = 3$.

**Double well (DW):** A typical problem in molecular dynamics considers sampling from the stationary distribution of a Langevin dynamics. In our example we consider a $d$-dimensional *double well* potential, corresponding to the (unnormalized) density $\rho(x) = \exp\left(-\sum_{i=1}^{m}(x_i^2 - \delta)^2 - \frac{1}{2}\sum_{i=m+1}^{d} x_i^2\right)$ with $m \in \mathbb{N}$ combined double wells (i.e., $2^m$ modes) and a separation parameter $\delta \in (0, \infty)$, see also Wu et al. (2020) and Figure 3. We choose a large value of $\delta$ to make sampling particularly challenging due to high energy barriers. Since $\rho$ factorizes in the dimensions, we obtain reference solutions by numerical integration and ground truth samples using rejection sampling with a Gaussian mixture proposal distribution, see also Midgley et al. (2023).

## 4.2 RESULTS

Let us start with the bridge approach and the general losses in (7) and (8). Table 5 (in the appendix) shows that the log-variance loss can improve our considered metrics significantly. However, the general bridge framework still suffers from reduced efficiency and numerical instabilities. For high-dimensional problems, it can be prohibitive to compute the divergence of $v$ using automatic

---

[3]The repository can be found at `https://github.com/juliusberner/sde_sampler`.

Table 2: PIS and DIS metrics for the benchmark problems of various dimensions $d$. We report the median over five independent runs, see Figure 9 for a corresponding boxplot. Specifically, we report errors for estimating the log-normalizing constant $\Delta \log Z$ as well the standard deviations $\Delta \operatorname{std}$ of the marginals. Furthermore, we report the normalized effective sample size ESS and the Sinkhorn distance $\mathcal{W}_\gamma^2$ (Cuturi, 2013), see Appendix A.6 for detailsMax-Joseph-Schule. The arrows $\uparrow$ and $\downarrow$ indicate whether we want to maximize or minimize a given metric.

| Problem | Method | Loss | $\Delta \log Z \downarrow$ | $\mathcal{W}_\gamma^2 \downarrow$ | ESS $\uparrow$ | $\Delta \operatorname{std} \downarrow$ |
|---|---|---|---|---|---|---|
| GMM ($d = 2$) | PIS | KL (Zhang & Chen, 2022) | 1.094 | 0.467 | 0.0051 | 1.937 |
| | | LV (ours) | **0.046** | **0.020** | **0.9093** | **0.023** |
| | DIS | KL (Berner et al., 2024) | 1.551 | 0.064 | 0.0226 | 2.522 |
| | | LV (ours) | **0.056** | **0.020** | **0.8660** | **0.004** |
| Funnel ($d = 10$) | PIS | KL (Zhang & Chen, 2022) | 0.288 | 5.639 | **0.1333** | 6.921 |
| | | LV (ours) | **0.277** | **5.593** | 0.0746 | **6.850** |
| | DIS | KL (Berner et al., 2024) | 0.433 | 5.120 | 0.1383 | 5.254 |
| | | LV (ours) | **0.430** | **5.062** | **0.2261** | **5.220** |
| DW ($d = 5, m = 5, \delta = 4$) | PIS | KL (Zhang & Chen, 2022) | 3.567 | 1.699 | 0.0004 | 1.409 |
| | | LV (ours) | **0.214** | **0.121** | **0.6744** | **0.001** |
| | DIS | KL (Berner et al., 2024) | 1.462 | 1.175 | 0.0012 | 0.431 |
| | | LV (ours) | **0.375** | **0.120** | **0.4519** | **0.001** |
| DW ($d = 50, m = 5, \delta = 2$) | PIS | KL (Zhang & Chen, 2022) | 0.101 | **6.821** | 0.8172 | 0.001 |
| | | LV (ours) | **0.087** | 6.823 | **0.8453** | **0.000** |
| | DIS | KL (Berner et al., 2024) | 1.785 | **6.854** | 0.0225 | 0.009 |
| | | LV (ours) | **1.783** | 6.855 | **0.0227** | 0.009 |

differentiation, and relying on Hutchinson's trace estimator introduces additional variance. We refer to Remark A.1 for further discussion. The instabilities might be rooted in the non-uniqueness of the optimal control (which follows from our analysis, cf. Section 3). Furthermore, such issues are also commonly observed in the context of SBs (De Bortoli et al., 2021; Chen et al., 2021a; Fernandes et al., 2021), where two controls need to be optimized. Therefore, for the more challenging problems, we focus on DIS and PIS, which do not incur the described pathologies.

We observe that the log-variance loss significantly improves both DIS and PIS across our considered benchmark problems and metrics, see Table 2. The improvements are especially remarkable considering that we only replaced the KL-based loss $\mathcal{L}_{\mathrm{KL}}$ by the log-variance loss $\mathcal{L}_{\mathrm{LV}}$ without tuning the hyperparameter for the latter loss. In the few cases where the KL divergence performs better, the difference seems rather insignificant. In particular, Figures 2 and 3 show that the log-variance loss successfully counteracts mode collapse, leading to quite substantial improvements. The benefit of the log-variance loss can also be observed for the benchmark posed in Wu et al. (2020), which aims to sample a target distribution resembling a picture of a Labrador, see Figure 8 in the appendix. In Appendix A.8, we present results for further (high-dimensional) targets, showing that diffusion-based samplers with log-variance loss are competitive with other state-of-the-art sampling methods.

## 5 CONCLUSION

In this work, we provide a unifying perspective on diffusion-based generative modeling that is based on path space measures of time-reversed diffusion processes and that, for the first time, connects methods such as SB, DIS, PIS, and DDS. Our novel framework also allows us to consider arbitrary divergences between path measures as objectives for the corresponding task of interest. While the KL divergence yields known methods, we find that choosing the log-variance divergence leads to novel algorithms that are particularly useful for the task of sampling from (unnormalized) densities. Specifically, this divergence exhibits beneficial properties, such as lower variance, computational efficiency, and exploration-exploitation trade-offs. We can demonstrate in multiple numerical examples that the log-variance loss greatly improves sampling quality across a range of metrics. We believe that problem and approach-specific finetuning might further enhance the performance of the log-variance loss, thereby paving the way for competitive diffusion-based sampling approaches.

ACKNOWLEDGMENTS

We thank Guan-Horng Liu for many useful discussions. The research of L.R. was funded by Deutsche Forschungsgemeinschaft (DFG) through the grant CRC 1114 "Scaling Cascades in Complex Systems" (project A05, project number 235221301). J.B. acknowledges support from the Wally Baer and Jeri Weiss Postdoctoral Fellowship.

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

# A  APPENDIX

## A.1  ASSUMPTIONS

In our proofs, we assume that the coefficient functions of all appearing SDEs are sufficiently regular such that Novikov's condition is satisfied and such that the SDEs admit unique strong solutions with smooth and strictly positive densities $p_{X_t}$ for $t \in (0, T)$, see, for instance, Arnold (1974); Øksendal & Øksendal (2003); Baldi (2017).

## A.2  PROOFS

*Proof of Proposition 2.3.* Let us define the path space measures $\mathbb{P}_{X^{u,x}}$ and $\mathbb{P}_{\overleftarrow{Y}^{v,x}}$ as the measures of $X^u$ and $\overleftarrow{Y}^v$ conditioned on $X_0^u = x$ and $\overleftarrow{Y}_0^v = x$ with $x \in \mathbb{R}^d$, respectively. We can then compute

$$\log \frac{\mathrm{d}\mathbb{P}_{X^u}}{\mathrm{d}\mathbb{P}_{\overleftarrow{Y}^v}}(X^w) = \log \frac{\mathrm{d}\mathbb{P}_{X^{u,x}}}{\mathrm{d}\mathbb{P}_{\overleftarrow{Y}^{v,x}}}(X^w) + \log \frac{\mathrm{d}\mathbb{P}_{X_0^u}}{\mathrm{d}\mathbb{P}_{\overleftarrow{Y}_0^v}}(X_0^w)$$

$$= \log \frac{\mathrm{d}\mathbb{P}_{X^{u,x}}}{\mathrm{d}\mathbb{P}_{\overleftarrow{Y}^{v,x}}}(X^w) + \log \frac{p_{\mathrm{prior}}(X_0^w)}{p_{\overleftarrow{Y}^v}(X_0^w, 0)}. \tag{16}$$

We follow Liu et al. (2022) and first note that the time-reversal of the process $Y^v$ defined in (3) is given by

$$\mathrm{d}\overleftarrow{Y}_s^v = (\mu + \sigma\sigma^\top\nabla g - \sigma v)(\overleftarrow{Y}_s^v, s)\,\mathrm{d}s + \sigma(s)\,\mathrm{d}W_s,$$

where we abbreviate $g := \log \bar{p}_{Y^v}$, see Lemma 2.1. Let us further define the short-hand notations $h := u + v - \sigma^\top\nabla g$ and $b := \mu + \sigma(u - h)$. Then, we can write the SDEs in (2) and (3) as

$$\begin{cases} \mathrm{d}X_s^u = (b + \sigma h)(X_s^u, s)\,\mathrm{d}s + \sigma(s)\,\mathrm{d}W_s, \\ \mathrm{d}\overleftarrow{Y}_s^v = b(\overleftarrow{Y}_s^v, s)\,\mathrm{d}s + \sigma(s)\,\mathrm{d}W_s. \end{cases}$$

We can now apply Girsanov's theorem (see, e.g., Nüsken & Richter, 2021, Lemma A.1) to rewrite the logarithm of the Radon-Nikodym derivative $\mathcal{R} := \log \frac{\mathrm{d}\mathbb{P}_{X^{u,x}}}{\mathrm{d}\mathbb{P}_{\overleftarrow{Y}^{v,x}}}(X^w)$ in (16) as

$$\mathcal{R} = \int_0^T \left(\sigma^{-\top} h\right)(X_s^w, s)\cdot\mathrm{d}X_s^w - \int_0^T \left(\sigma^{-1}b\cdot h\right)(X_s^w, s)\,\mathrm{d}s - \frac{1}{2}\int_0^T \|h(X_s^w, s)\|^2\,\mathrm{d}s$$

$$= \int_0^T \left((w - u)\cdot h + \frac{1}{2}\|h\|^2\right)(X_s^w, s)\,\mathrm{d}s + S_h(X^w)$$

$$= \int_0^T \left((w - u)\cdot\left(u + v - \sigma^\top\nabla g\right) + \frac{1}{2}\|u + v - \sigma^\top\nabla g\|^2\right)(X_s^w, s)\,\mathrm{d}s + S_h(X^w) \tag{17}$$

$$= R_{f_{u,v,w}^{\mathrm{Bridge}}} - \int_0^T \left(\nabla\cdot(\sigma v - \mu) + (v + w)\cdot\sigma^\top\nabla g - \frac{1}{2}\|\sigma^\top\nabla g\|^2\right)(X_s^w, s)\,\mathrm{d}s + S_h(X^w).$$

Further, we may apply Itô's lemma to the function $g$ to get

$$g(X_T^w, T) - g(X_0^w, 0) = \int_0^T \left(\partial_s g + \nabla g\cdot(\mu + \sigma w) + \frac{1}{2}\mathrm{Tr}\left(\sigma\sigma^\top\nabla^2 g\right)\right)(X_s^w, s)\,\mathrm{d}s + S_{\sigma^\top\nabla g}(X^w).$$

Noting that $g = \log p_{\overleftarrow{Y}^v}$ fulfills the Hamilton-Jacobi-Bellman equation (see, e.g., Berner et al., 2024)

$$\partial_s g = -\frac{1}{2}\mathrm{Tr}\left(\sigma\sigma^\top\nabla^2 g\right) + (\sigma v - \mu)\cdot\nabla g + \nabla\cdot(\sigma v - \mu) - \frac{1}{2}\|\sigma^\top g\|^2,$$

we get

$$g(X_T^w, T) - g(X_0^w, 0) = \int_0^T \left(\nabla\cdot(\sigma v - \mu) + (v + w)\cdot\sigma^\top\nabla g - \frac{1}{2}\|\sigma^\top g\|^2\right)(X_s^w, s) + S_{\sigma^\top\nabla g}(X^w).$$

Finally, combining this with (16) and (17) and noting that

$$g(X_T^w, T) = \log p_{\overleftarrow{Y}^v}(X_T^w, T) = \log p_{Y^v}(X_T^w, 0) = p_{\mathrm{target}}(X_T^w),$$

yields the desired expression. $\qquad\qquad\square$

**Remark A.1** (Divergence-free objectives). One can remove the divergence from the Radon-Nikodym derivative (6) and thus from corresponding losses by noting the identity

$$\int_0^T \nabla \cdot (\sigma v - \mu)(X_s^w, s) \, \mathrm{d}s = \int_0^T \left( v - \sigma^{-1}\mu \right)(X_s^w, s) \cdot \left( \overleftarrow{\mathrm{d}}W_s - \mathrm{d}W_s \right),$$

where for a suitable function $\varphi \in C(\mathbb{R}^d \times [0,T], \mathbb{R}^d)$ the backward integration w.r.t. Brownian motion is defined as

$$\int_0^T \varphi(X_s^w, s) \cdot \overleftarrow{\mathrm{d}}W_s := \lim_{\Delta t \to 0} \sum_{n=1}^N \varphi(X_{t_{n+1}}^w, t_{n+1}) \cdot \left( W_{t_{n+1}} - W_{t_n} \right),$$

which, in contrast to the definition of the usual Itô integral,

$$\int_0^T \varphi(X_s^w, s) \cdot \mathrm{d}W_s := \lim_{\Delta t \to 0} \sum_{n=1}^N \varphi(X_{t_n}^w, t_n) \cdot \left( W_{t_{n+1}} - W_{t_n} \right),$$

considers the right endpoint when discretizing the integral on a time grid $0 = t_0 < t_1 < \cdots < t_N = T$ with step size $\Delta t := t_{n+1} - t_n$. The above definitions via refined partitions readily bring implementation schemes for both integrals when choosing a fixed step size $\Delta t > 0$. Divergence-free objectives might be particularly beneficial in higher dimensions, where it is typically expensive to compute the divergence using automatic differentiation. For further details, we refer to Vargas et al. (2024) and Kunita (2019).

*Proof of Proposition 2.5.* Let us first recall the notion of Gâteaux derivatives, see Siddiqi & Nanda (1986, Section 5.2). We say that $\mathcal{L} \colon \mathcal{U} \times \mathcal{U} \to \mathbb{R}_{\geq 0}$ is *Gâteaux differentiable* at $u \in \mathcal{U}$ if for all $v, \phi \in \mathcal{U}$ the mapping

$$\varepsilon \mapsto \mathcal{L}(u + \varepsilon\phi, v)$$

is differentiable at $\varepsilon = 0$. The Gâteaux derivative of $\mathcal{L}$ w.r.t. $u$ in direction $\phi$ is then defined as

$$\frac{\delta}{\delta u}\mathcal{L}(u, v; \phi) := \frac{\mathrm{d}}{\mathrm{d}\varepsilon}\Big|_{\varepsilon=0} \mathcal{L}(u + \varepsilon\phi, v).$$

The derivative of $\mathcal{L}$ w.r.t. $v$ is defined analogously. Let now $u = u_\theta$ and $v = v_\gamma$ be parametrized[4] by $\theta \in \mathbb{R}^p$ and $\gamma \in \mathbb{R}^p$. Relating the Gâteaux derivatives to partial derivatives w.r.t. $\theta$ and $\gamma$, respectively, let us note that we are particularly interested in the directions $\phi = \partial_{\theta_i} u_\theta$ and $\phi = \partial_{\gamma_i} v_\gamma$ for $i \in \{1, \ldots, p\}$. This choice is motivated by the chain rule of the Gâteaux derivative, which, under suitable assumptions, states that

$$\partial_{\theta_i}\mathcal{L}(u_\theta, v_\gamma) = \frac{\delta}{\delta u}\Big|_{u=u_\theta} \mathcal{L}(u, v_\gamma; \partial_{\theta_i} u_\theta) \quad \text{and} \quad \partial_{\gamma_i}\mathcal{L}(u_\theta, v_\gamma) = \frac{\delta}{\delta v}\Big|_{v=v_\gamma} \mathcal{L}(u_\theta, v; \partial_{\gamma_i} v_\gamma).$$

Analogous to the computations in Nüsken & Richter (2021), the Gâteaux derivatives of the Monte Carlo estimator $\widehat{\mathcal{L}}_{\mathrm{LV}}^w$ of the log-variance loss $\mathcal{L}_{\mathrm{LV}}^w$ in (8) with $K \in \mathbb{N}$ samples is given by

$$\frac{\delta}{\delta u}\widehat{\mathcal{L}}_{\mathrm{LV}}^w(u, v; \phi) = \frac{2}{K}\sum_{k=1}^K \mathcal{A}_{u,v,w}^{(k)} \left( \mathcal{B}_{u,w,\phi}^{(k)} - \frac{1}{K}\sum_{i=1}^K \mathcal{B}_{u,w,\phi}^{(i)} \right), \tag{18}$$

where the superscript $(k)$ denotes the index of the $k$-th i.i.d. sample in the Monte Carlo estimator $\widehat{\mathcal{L}}_{\mathrm{LV}}^w$ and we define the short-hand notations

$$\mathcal{A}_{u,v,w}^{(k)} := \left( R_{f_{u,v,w}^{\mathrm{Bridge}}} + S_{u+v}^{(k)} + B \right)(X^{w,(k)}) + \log Z$$

and

$$\mathcal{B}_{u,w,\phi}^{(k)} := \left( R_{f_{u,w,\phi}^{\mathrm{gen}}} + S_\phi^{(k)} \right)(X^{w,(k)}) \quad \text{with} \quad f_{u,w,\phi}^{\mathrm{gen}} = (w - u) \cdot \phi.$$

Now, note that the definition of the log-variance loss and Proposition 2.3 imply that for the optimal choices $u = u^*, v = v^*$ it holds that

$$\mathcal{A}_{u^*,v^*,w}^{(k)} = 0$$

---

[4] We only assume that $\theta$ and $\gamma$ are in the same space $\mathbb{R}^p$ for notational simplicity.

almost surely for every $k \in \{1, \ldots, K\}$ and $w \in \mathcal{U}$. This readily implies the statement for the derivative w.r.t. the control $u_\gamma$. The analogous statement holds true for the derivative w.r.t. $v_\gamma$, as we can compute

$$\frac{\delta}{\delta v} \widehat{\mathcal{L}}_{\mathrm{LV}}^w(u, v; \phi) = \frac{2}{K} \sum_{k=1}^{K} \mathcal{A}_{u,v,w}^{(k)} \left( \mathcal{C}_{v,w,\phi}^{(k)} - \frac{1}{K} \sum_{i=1}^{K} \mathcal{C}_{v,w,\phi}^{(i)} \right),$$

where

$$\mathcal{C}_{v,w,\phi}^{(k)} = \left( R_{f_{v,w,\phi}^{\inf}} + S_\phi^{(k)} \right)(X^{w,(k)}) \quad \text{with} \quad f_{v,w,\phi}^{\inf} = (v + w) \cdot \phi + \nabla \cdot (\sigma \phi).$$

For the derivative of the Monte Carlo version of the loss $\mathcal{L}_{\mathrm{KL}}$ as defined in (7) w.r.t. to $v$ we may compute

$$\frac{\delta}{\delta v} \widehat{\mathcal{L}}_{\mathrm{KL}}(u, v; \phi) = \frac{1}{K} \sum_{k=1}^{K} \int_0^T ((u + v) \cdot \phi + \nabla \cdot (\sigma \phi))(X_s^{u,(k)}, s) \, \mathrm{d}s.$$

We note that even for $u = u^*$ and $v = v^*$ we can usually not expect the variance of the corresponding Monte Carlo estimator to be zero. For the computation of the derivative w.r.t. $u$ we refer to Nüsken & Richter (2021, Proposition 5.3). □

**Remark A.2** (Control variate interpretation). For the gradient of the loss $\mathcal{L}_{\mathrm{KL}}$ w.r.t. to $u$ we may compute

$$\frac{\delta}{\delta u} \mathcal{L}_{\mathrm{KL}}(u, v; \phi) = \mathbb{E}\left[ \int_0^T ((u + v) \cdot \phi)(X_s^u, s) \, \mathrm{d}s + \left( R_{f_{u,v,u}^{\mathrm{Bridge}}}(X^u) + B(X^u) \right) S_\phi(X^u) \right]$$

$$= \mathbb{E}\left[ \mathcal{A}_{u,v,u} S_\phi(X^u) \right],$$

where we used Girsanov's theorem and the Itô isometry. Comparing with (18), we realize that the derivative of $\mathcal{L}_{\mathrm{LV}}$ w.r.t. $u$ for the choice $w = u$ can be interpreted as a control variate version of the derivative of $\mathcal{L}_{\mathrm{KL}}$, thereby promising reduced variance of the corresponding Monte Carlo estimators, cf. Nüsken & Richter (2021); Richter et al. (2020).

In the context of reinforcement learning, such a control variate is also known as *local baseline*. As an alternative, *global baselines* have been proposed, where the batch-dependent scaling of the local baseline is replaced by an exponentially moving average. This corresponds to replacing the variance in the loss with the second moment and additionally optimizing an approximation of the log-normalizing constant (with a specific learning rate), see Malkin et al. (2022b). The resulting loss is then known as (second) moment loss (Nüsken & Richter, 2021; Richter et al., 2020) or trajectory balance objective (Malkin et al., 2022a).

**Remark A.3** (Alternative derivations of Lemma 3.1). The expression in Lemma 3.1 can also be derived via

$$\frac{\mathrm{d}\mathbb{P}_{X^u}}{\mathrm{d}\mathbb{P}_{\bar{Y}^{\bar{v}}}}(X^w) = \frac{\mathrm{d}\mathbb{P}_{X^u}}{\mathrm{d}\mathbb{P}_{X^r}}(X^w) \frac{\mathrm{d}\mathbb{P}_{\bar{Y}^{\bar{v},\mathrm{ref}}}}{\mathrm{d}\mathbb{P}_{\bar{Y}^{\bar{v}}}}(X^w) = \frac{\mathrm{d}\mathbb{P}_{X^u}}{\mathrm{d}\mathbb{P}_{X^r}}(X^w) \frac{p_{X_T^r}}{p_{\mathrm{target}}}(X_T^w),$$

where the first factor can be computed via recalling

$$\frac{\mathrm{d}\mathbb{P}_{X^u}}{\mathrm{d}\mathbb{P}_{X^r}}(X^w) = \exp\left( R_{f_{u,r,w}^{\mathrm{ref}}} + S_{u-r} \right)(X^w). \tag{19}$$

Yet another viewpoint is based on importance sampling in path space, see, e.g., Hartmann et al. (2017). Since our goal is to find an optimal control $u^*$ such that we get samples $X_T^{u^*} \sim p_{\mathrm{target}}$, we may define our target path space measure $\mathbb{P}_{X^{u^*}}$ via $\frac{\mathrm{d}\mathbb{P}_{X^{u^*}}}{\mathrm{d}\mathbb{P}_{X^r}}(X^w) = \frac{p_{\mathrm{target}}}{p_{X_T^r}}(X_T^w)$. We can then compute

$$\frac{\mathrm{d}\mathbb{P}_{X^u}}{\mathrm{d}\mathbb{P}_{X^{u^*}}}(X^w) = \frac{\mathrm{d}\mathbb{P}_{X^u}}{\mathrm{d}\mathbb{P}_{X^r}}(X^w) \frac{\mathrm{d}\mathbb{P}_{X^r}}{\mathrm{d}\mathbb{P}_{X^{u^*}}}(X^w),$$

which, together with (19), is equivalent to the expression in Lemma 3.1. Note that in the importance sampling perspective we do not need the concept of time-reversals.

### A.3 Sampling via learned diffusions

In the following, we provide a high-level overview of sampling methods based on controlled diffusion processes. We base our explanation on the general KL-based loss stated in (7) since most previous methods are special cases of this formulation, see Table 1.

Let us recall that we want to learn a control $u$ such that $X_T^u \sim p_{\text{target}}$. We first observe that the terminal costs $B(X^u)$ or $B^{\text{ref}}(X^u)$ contain the term $-\log \rho = -\log p_{\text{target}} + \log Z$, which penalizes $X_T^u$ for ending up at regions with low probability w.r.t. the target density. The other terms of the terminal cost, together with the running costs $R_{f_{u,v,\tilde{u}}^{\text{Bridge}}}$, are enforcing additional constraints on the trajectories of our process $X^u$. In our formulation, they generally enforce $X^u$ to be the time-reversal of $Y^v$. For special choices of $v$, this yields the following settings:

- For the PIS method, we minimize the reverse KL divergence of the controlled process $X^u$ to the uncontrolled process $X^0$, promoting $u$ to be as close to zero as possible. This corresponds to a classical Schrödinger bridge problem, see Appendix A.4, which, for the simple initial condition $X_0^0 \sim \delta_{x_0}$, can be solved without sequential optimization routines, see also Section 1.1 and Appendix A.4.1.
- The DIS and DDS methods are motivated by diffusion-based generative modeling (Ho et al., 2020; Kingma et al., 2021; Nichol & Dhariwal, 2021; Vahdat et al., 2021; Song & Ermon, 2020). In particular, they minimize the reverse KL divergence to the time-reversed "noising" process $Y^0$. In other words, $X^u$ is enforced to denoise the samples $Y_T^0$ in order to yield samples from $Y_0^0 \sim p_{\text{target}}$.

While we base our unifying framework in Section 2 on the perspective of path measures, the respective methods for the KL divergence can also be derived from the underlying PDEs or BSDE systems, see Appendix A.4 and Berner et al. (2024).

### A.4 The Schrödinger bridge problem

In this section, we provide some background information on the classical Schrödinger bridge problem. Recall from Section 3.1 that out of all solutions $u^*$ fulfilling the general bridge problem stated in Problem 2.2, which can be characterized by Nelson's identity in (9), the *Schrödinger bridge problem* considers the solution $u^*$ that minimizes the KL divergence $D_{\text{KL}}(\mathbb{P}_{X^{u^*}} | \mathbb{P}_{X^r})$ to a given reference process $X^r$, defined as in (2) with $u$ replaced by $r \in \mathcal{U}$, i.e.

$$\mathrm{d}X_s^r = (\mu + \sigma r)(X_s^r, s)\,\mathrm{d}s + \sigma(s)\,\mathrm{d}W_s, \qquad X_0^r \sim p_{\text{prior}}.$$

Traditionally, the uncontrolled process $X^0$ with $r = 0$ is chosen, i.e.,

$$\mathrm{d}X_s^0 = \mu(X_s^0, s)\,\mathrm{d}s + \sigma(s)\,\mathrm{d}W_s, \qquad X_0^0 \sim p_{\text{prior}}.$$

In the following, we will formulate optimality conditions for the Schrödinger bridge problem defined in (12) for this standard case $r = 0$. Moreover, we outline how the associated BSDE system leads to the same losses as given in (7) and (8), respectively. The ideas are based on Chen et al. (2021a); Vargas (2021); Liu et al. (2022); Caluya & Halder (2021).

First, we can define the

$$\phi(x, t) := \min_{u \in \mathcal{U}} \mathbb{E}\left[\frac{1}{2}\int_t^T \|u(X_s^u, s)\|^2\,\mathrm{d}s \,\middle|\, X_t^u = x,\ X_T^u \sim p_{\text{target}}\right].$$

By the *dynamic programming principle* it holds that $\phi$ solves the *Hamilton-Jacobi-Bellman* (HJB) equation

$$\partial_t \phi = -\mu \cdot \nabla \phi - \frac{1}{2}\operatorname{Tr}\left(\sigma\sigma^\top \nabla^2 \phi\right) + \frac{1}{2}\left\|\sigma^\top \nabla \phi\right\|^2 \tag{20}$$

(with unknown boundary conditions) and that the optimal control satisfies

$$u^* = -\sigma^\top \nabla \phi.$$

Together with the corresponding Fokker-Planck equation for $X^{u^*}$, this yields necessary and sufficient conditions for the solution to (11). Now, we can transform the Fokker-Planck equation and the HJB equation (20) into a system of linear equations, using the exponential transform

$$\psi := \exp(-\phi) \quad \text{and} \quad \widehat{\psi} := p_{X^{u^*}} \exp(\phi) = \frac{p_{X^{u^*}}}{\psi}, \tag{21}$$

often referred to as the *Hopf-Cole transform*. This yields the following well-known optimality conditions of the Schrödinger bridge problem defined in (12).

**Theorem A.4** (Optimality PDEs). *The solution $u^*$ to the Schrödinger bridge problem* (12) *is equivalently given by*

1. $u^* := -\sigma^\top \nabla \phi$, *where $p_{X^{u^*}}$ and $\phi$ are the unique solutions to the coupled PDEs*

$$\begin{cases} \partial_t p_{X^{u^*}} = -\nabla \cdot \left( p_{X^{u^*}} (\mu - \sigma \sigma^\top \nabla \phi) \right) + \frac{1}{2} \operatorname{Tr} \left( \sigma \sigma^\top \nabla^2 p_{X^{u^*}} \right) \\ \partial_t \phi = -\mu \cdot \nabla \phi - \frac{1}{2} \operatorname{Tr} \left( \sigma \sigma^\top \nabla^2 \phi \right) + \frac{1}{2} \left\| \sigma^\top \nabla \phi \right\|^2, \end{cases}$$

*with boundary conditions*

$$\begin{cases} p_{X^{u^*}}(\cdot, 0) = p_{\text{prior}}, \\ p_{X^{u^*}}(\cdot, T) = p_{\text{target}}. \end{cases}$$

2. $u^* := \sigma^\top \nabla \log \psi$, *where $\psi$ and $\widehat{\psi}$ are the unique solutions to the PDEs*

$$\begin{cases} \partial_t \psi = -\nabla \psi \cdot \mu - \frac{1}{2} \operatorname{Tr} \left( \sigma \sigma^\top \nabla^2 \psi \right), \\ \partial_t \widehat{\psi} = -\nabla \cdot \left( \widehat{\psi} \mu \right) + \frac{1}{2} \operatorname{Tr} \left( \sigma \sigma^\top \nabla^2 \widehat{\psi} \right), \end{cases} \tag{22}$$

*with coupled boundary conditions*

$$\begin{cases} \psi(\cdot, 0) \widehat{\psi}(\cdot, 0) = p_{\text{prior}}, \\ \psi(\cdot, T) \widehat{\psi}(\cdot, T) = p_{\text{target}}. \end{cases} \tag{23}$$

*The optimal control $v^*$ is given by Nelson's identity* (9), *i.e.,*

$$v^* = \sigma^\top \nabla \log p_{X^{u^*}} - u^* = \sigma^\top \nabla \log \widehat{\psi}. \tag{24}$$

Using Itô's lemma, we now derive a BSDE system corresponding to the PDE system in (22).

**Proposition A.5** (BSDEs for the SB problem). *Let us assume $\psi$ and $\widehat{\psi}$ fulfill the PDEs* (22) *with boundary conditions* (23) *and let us define the processes*

$$\begin{cases} \mathcal{Y}_s^w = \log \psi(X_s^w, s), \\ \widehat{\mathcal{Y}}_s^w = \log \widehat{\psi}(X_s^w, s), \\ \mathcal{Z}_s^w = \sigma^\top \nabla \log \psi(X_s^w, s) = u^*(X_s^w, s), \\ \widehat{\mathcal{Z}}_s^w = \sigma^\top \nabla \log \widehat{\psi}(X_s^w, s) = v^*(X_s^w, s), \end{cases}$$

*where the process $X^w$ is given by*

$$\mathrm{d}X_s^w = (\mu + \sigma w)(X_s^w, s) \, \mathrm{d}s + \sigma(s) \, \mathrm{d}W_s$$

*with $w \in \mathcal{U}$ being an arbitrary control function. We then get the BSDE system*

$$\begin{cases} \mathrm{d}\mathcal{Y}_s^w = \left( \mathcal{Z}_s^w \cdot w(X_s^w, s) - \frac{1}{2} \|\mathcal{Z}_s^w\|^2 \right) \mathrm{d}s + \mathcal{Z}_s^w \cdot \mathrm{d}W_s, \\ \mathrm{d}\widehat{\mathcal{Y}}_s^w = \left( \frac{1}{2} \|\widehat{\mathcal{Z}}_s^w\|^2 + \nabla \cdot (\sigma \widehat{\mathcal{Z}}_s^w - \mu(X_s^w, s)) + \widehat{\mathcal{Z}}_s^w \cdot w(X_s^w, s) \right) \mathrm{d}s + \widehat{\mathcal{Z}}_s^w \cdot \mathrm{d}W_s. \end{cases}$$

*Furthermore, it holds*

$$\mathcal{Y}_s^w + \widehat{\mathcal{Y}}_s^w = \log p_{X^{u^*}}(X_s^w, s) = \log \breve{p}_{Y^{v^*}}(X_s^w, s). \tag{25}$$

*Proof.* The proof is similar to the one in Chen et al. (2021a). For brevity, we define $D = \frac{1}{2} \sigma \sigma^\top$. We can apply Itô's lemma to the stochastic process $\mathcal{Y}_s^w = \log \psi(X_s^w, s)$ and get

$$\mathrm{d}\mathcal{Y}_s^w = \left( \partial_s \log \psi + \nabla \log \psi \cdot (\mu + \sigma w) + \operatorname{Tr} \left( D \nabla^2 \log \psi \right) \right) (X_s^w, s) \, \mathrm{d}s + \sigma^\top \nabla \log \psi(X_s^w, s) \cdot \mathrm{d}W_s.$$

Further, via (22) it holds

$$\partial_s \log \psi = \frac{1}{\psi} \left( -\nabla \psi \cdot \mu - \operatorname{Tr} \left( D \nabla^2 \psi \right) \right) = -\nabla \log \psi \cdot \mu - \operatorname{Tr} \left( \frac{D \nabla^2 \psi}{\psi} \right),$$

and we note the identity

$$\nabla^2 \log \psi = \frac{\nabla^2 \psi}{\psi} - \frac{\nabla \psi \, (\nabla \psi)^\top}{\psi^2}.$$

Combining the previous three equations, we get

$$
\begin{aligned}
\mathrm{d}\mathcal{Y}_s^w &= \left( \sigma^\top \nabla \log \psi \cdot w - \mathrm{Tr}\left( D \frac{\nabla \psi \, (\nabla \psi)^\top}{\psi^2} \right) \right)(X_s^w, s) \, \mathrm{d}s + \sigma^\top \nabla \log \psi (X_s^w, s) \cdot \mathrm{d}W_s \\
&= \left( \mathcal{Z}_s^w \cdot w(X_s^w, s) - \frac{1}{2}\|\mathcal{Z}_s^w\|^2 \right) \mathrm{d}s + \mathcal{Z}_s^w \cdot \mathrm{d}W_s.
\end{aligned}
$$

Similarly, we may apply Itô's lemma to $\widehat{\mathcal{Y}}_s^w = \log \widehat{\psi}(X_s^w, s)$ and get

$$\mathrm{d}\widehat{\mathcal{Y}}_s^w = \left( \partial_s \log \widehat{\psi} + \nabla \log \widehat{\psi} \cdot (\mu + \sigma w) + \mathrm{Tr}\left( D \nabla^2 \log \widehat{\psi} \right) \right)(X_s^w, s)\, \mathrm{d}s + \widehat{\mathcal{Z}}_s^w \cdot \mathrm{d}W_s.$$

Now, via (22) it holds that

$$\partial_s \log \widehat{\psi} = \frac{1}{\widehat{\psi}}\left( -\nabla \cdot \left( \widehat{\psi}\mu \right) + \mathrm{Tr}\left( D\nabla^2 \widehat{\psi} \right) \right) = -\nabla \log \widehat{\psi} \cdot \mu - \nabla \cdot \mu + \mathrm{Tr}\left( \frac{D\nabla^2 \widehat{\psi}}{\widehat{\psi}} \right).$$

Combining the previous two equations, we get

$$\mathrm{d}\widehat{\mathcal{Y}}_s^w = \left( \mathrm{Tr}\left( D\frac{\nabla^2 \widehat{\psi}}{\widehat{\psi}} + D\nabla^2 \log \widehat{\psi} \right) - \nabla \cdot \mu + \sigma^\top \nabla \log \widehat{\psi} \cdot w \right)(X_s^w, s)\, \mathrm{d}s + \widehat{\mathcal{Z}}_s^w \cdot \mathrm{d}W_s.$$

Now, noting the identity

$$
\begin{aligned}
\mathrm{Tr}\left( D\frac{\nabla^2 \widehat{\psi}}{\widehat{\psi}} + D\nabla^2 \log \widehat{\psi} \right) &= 2\,\mathrm{Tr}\left( D\frac{\nabla^2 \widehat{\psi}}{\widehat{\psi}} \right) - \frac{1}{2}\|\sigma^\top \nabla \log \widehat{\psi}\|^2 \\
&= \frac{1}{2}\|\sigma^\top \nabla \log \widehat{\psi}\|^2 + \nabla \cdot \left( \sigma\sigma^\top \nabla \log \widehat{\psi} \right),
\end{aligned}
$$

we can get the relation

$$
\begin{aligned}
\mathrm{d}\widehat{\mathcal{Y}}_s^w &= \left( \frac{1}{2}\|\sigma^\top \nabla \log \widehat{\psi}\|^2 + \nabla \cdot \left( \sigma\sigma^\top \nabla \log \widehat{\psi} - \mu \right) + \sigma^\top \nabla \log \widehat{\psi} \cdot w \right)(X_s^w, s)\, \mathrm{d}s + \widehat{\mathcal{Z}}_s^w \cdot \mathrm{d}W_s \\
&= \left( \frac{1}{2}\|\widehat{\mathcal{Z}}_s^w\|^2 + \nabla \cdot (\sigma\widehat{\mathcal{Z}}_s^w - \mu) + \widehat{\mathcal{Z}}_s^w \cdot w \right)(X_s^w, s)\, \mathrm{d}s + \widehat{\mathcal{Z}}_s^w \cdot \mathrm{d}W_s,
\end{aligned}
$$

which concludes the proof. $\qquad\square$

Note that the BSDE system is slightly more general than the one introduced in Chen et al. (2021a), which can be recovered with the choice $w(X_s^w, s) = \mathcal{Z}_s^w$. Also, the roles of $p_{\mathrm{prior}}$ and $p_{\mathrm{target}}$ are interchanged in Chen et al. (2021a) since they consider generative modeling instead of sampling from densities.

A valid loss can now be derived by adding the two BSDEs and recalling relation (25), which yields

$$
\begin{aligned}
-B(X^w) - \log(Z) = \log \frac{p_{\mathrm{target}}(X_T^w)}{p_{\mathrm{prior}}(X_0^w)} &= \left( \mathcal{Y}_T^w + \widehat{\mathcal{Y}}_T^w \right) - \left( \mathcal{Y}_0^w + \widehat{\mathcal{Y}}_0^w \right) \\
&= \left( R_{f_{u^*, v^*, w}^{\mathrm{Bridge}}} + S_{u^* + v^*} \right)(X^w)
\end{aligned}
$$

almost surely. Analogous to Berner et al. (2024); Huang et al. (2021) in generative modeling, the above equality suggests a parameterized lower bound of the log-likelihood $\log p_{\mathrm{prior}}$ when replacing the optimal controls in $\mathcal{Z}_s^w = u^*(X^w, s)$ and $\widehat{\mathcal{Z}}_s^w = v^*(X^w, s)$ with their approximations $u$ and $v$, see Chen et al. (2021a). This lower bound exactly recovers the loss given in (7). Further, note that the variance of the left-hand minus the right-hand side is zero, which readily yields our log-variance loss as defined in (8).

### A.4.1 SCHRÖDINGER HALF-BRIDGES (PIS)

For the Schrödinger half-bridge, also referred to as PIS, introduced in Section 3.3, we can find an alternative derivation, motivated by the PDE perspective outlined in Appendix A.4. For this derivation it is crucial that we assume the prior density to be concentrated at a single point, i.e., $p_{\text{prior}} := \delta_{x_0}$ for some $x_0 \in \mathbb{R}^d$ (typically $x_0 = 0$), see Tzen & Raginsky (2019); Dai Pra (1991). We can recover the corresponding objectives by noting that, in the case $p_{\text{prior}} = \delta_{x_0}$, the system of PDEs in (22) can be decoupled. More precisely, we observe that the second equation in (22) is the Fokker-Planck equation of $X^0$ and we have that

$$\widehat{\psi} = p_{X^{u*}} \exp(\phi) = p_{X^0} \quad \text{and} \quad \widehat{\psi}(\cdot, 0) = p_{X_0^0} = \delta_{x_0}.$$

In view of (24), we note that this defines $v^* = \sigma^\top \nabla \log p_{X^0}$. By (21), we observe that $\psi = \frac{p_{X^{u*}}}{p_{X^0}}$, which yields the boundary condition

$$\phi(\cdot, T) = -\log \psi(\cdot, T) = \log \frac{p_{X_T^0}}{p_{\text{target}}} = \log \frac{Z p_{X_T^0}}{\rho}$$

to the HJB equation in (20). By the *verification theorem* (Dai Pra, 1991; Pavon, 1989; Nüsken & Richter, 2021; Fleming & Soner, 2006; Pham, 2009), we thus obtain the PIS objective

$$\mathcal{L}_{\text{KL}}(u) = \mathbb{E}\left[\frac{1}{2}\int_0^T \|u(X_s^u, s)\|^2 \, \mathrm{d}s + \log \frac{p_{X_T^0}(X_T^u)}{\rho(X_T^u)}\right] = \mathbb{E}\left[\left(R_{f_{u,0,u}^{\text{ref}}} + B^{\text{ref}}\right)(X^u)\right].$$

Moreover, the optimal control is given by $u^* = -\sigma^\top \nabla \phi = \sigma^\top \nabla \log \psi$. We can also derive this objective from the BSDE system in Proposition A.5. Since $\widehat{\psi}(\cdot, 0) = \delta_{x_0}$, we may focus on the process $\mathcal{Y}_s^w = \log \psi(X_s^w, s)$ only, and get

$$\mathcal{Y}_T^w - \mathcal{Y}_0^w = \int_0^T \mathcal{Z}_s^w \cdot w(X_s^w, s) - \frac{1}{2}\|\mathcal{Z}_s^w\|^2 \, \mathrm{d}s + \int_0^T \mathcal{Z}_s^w \cdot \mathrm{d}W_s.$$

The PIS objective now follows by choosing $w(X_s^w, s) = \mathcal{Z}_s^w$ and noting that

$$\mathcal{Y}_T^w = \log \psi(X_T^w, T) = \log \frac{p_{\text{target}}}{p_{X_T^0}}(X_T^w).$$

Recalling our notation in (1), this also shows that the log-variance loss can be written as

$$\mathcal{L}_{\text{LV}}^w(u) = \mathbb{V}\left[\left(R_{f_{u,0,w}^{\text{ref}}} + S_u + B^{\text{ref}}\right)(X^w)\right].$$

### A.5 TRACTABLE SDEs

Let us present some commonly used SDEs of the form

$$\mathrm{d}X_s^u = \mu(X_s^u, s) \, \mathrm{d}s + \sigma(s) \, \mathrm{d}W_s$$

with affine drifts that have tractable marginals conditioned on their initial value, see Song et al. (2020). For notational convenience, let us define

$$\alpha(t) := \int_0^t \beta(s)\mathrm{d}s$$

with suitable $\beta \in C([0, T], (0, \infty))$.

**Variance-preserving (VP) SDE:** This *Ornstein-Uhlenbeck* process is given by

$$\sigma(t) := \nu\sqrt{2\beta(t)}\,\mathrm{I} \quad \text{and} \quad \mu(x, t) := -\beta(t)x.$$

with $\nu \in (0, \infty)$. Then, we have that

$$X_t | X_0 \sim \mathcal{N}\left(e^{-\alpha(t)}X_0, \nu^2\left(1 - e^{-2\alpha(t)}\right)\mathrm{I}\right).$$

This shows that for $\alpha(T)$ sufficiently large it holds that $X_T \approx \mathcal{N}\left(0, \nu^2\mathrm{I}\right)$. For $X_0 \sim \mathcal{N}(m, \Sigma)$, we further have that

$$X_t \sim \mathcal{N}\left(e^{-\alpha(t)}m, e^{-2\alpha(t)}\left(\Sigma - \nu^2\mathrm{I}\right) + \nu^2\mathrm{I}\right). \tag{26}$$

---

**Algorithm 1** Training of a generalized time-reversed diffusion sampler

---

**Input:** neural networks $u_\theta, v_\gamma$ with initial parameters $\theta, \gamma$, optimizer method $\mathrm{step}$ for updating the parameters, number of steps $K$, batch size $m$

**Output:** optimized parameters $\theta, \gamma$

> **for** $k \leftarrow 1, \ldots, K$ **do**
>> $\mathcal{L} \leftarrow$ choose KL-based loss $\mathcal{L}_{\mathrm{KL}}$ in (7) or log-variance loss $\mathcal{L}_{\mathrm{LV}}$ in (8)      ▷ Setup
>> **if** $\mathcal{L} = \mathcal{L}_{\mathrm{KL}}$ **then**
>>> $w \leftarrow u_\theta$
>>> $p \leftarrow p_{\mathrm{prior}}$
>> **else**
>>> $w \leftarrow$ choose (detached) control for the forward process
>>> $p \leftarrow$ choose initial distribution for the forward process
>> **end if**
>>
>> **for** $i = 1, \ldots, m$ **do**      ▷ Approximate cost (batched in practice)
>>> $x \leftarrow$ sample from $p$
>>> $(W, X^w) \leftarrow$ simulate discretizations of Brownian motion $W$ and SDE $X^w$ with $X_0^w = x$
>>> $(R_{f_{u_\theta, v_\gamma, w}^{\mathrm{Bridge}}}, B) \leftarrow$ compute approximations of the running and terminal costs using $X^w$
>>> $\mathrm{rnd}_i \leftarrow R_{f_{u_\theta, v_\gamma, w}^{\mathrm{Bridge}}} + B$
>>> **if** $\mathcal{L} = \mathcal{L}_{\mathrm{LV}}$ **then**
>>>> $S_{u_\theta + v_\gamma} \leftarrow$ compute approximation of the stochastic integral using $W$
>>>> $\mathrm{rnd}_i \leftarrow \mathrm{rnd}_i + S_{u_\theta + v_\gamma}$
>>> **end if**
>> **end for**
>>
>> $\mathrm{mean} \leftarrow \frac{1}{m} \sum_{i=1}^{m} \mathrm{rnd}_i$      ▷ Compute loss
>> **if** $\mathcal{L} = \mathcal{L}_{\mathrm{KL}}$ **then**
>>> $\widehat{\mathcal{L}} \leftarrow \mathrm{mean}$
>> **else**
>>> $\widehat{\mathcal{L}} \leftarrow \frac{1}{m-1} \sum_{i=1}^{m} (\mathrm{rnd}_i - \mathrm{mean})^2$
>> **end if**
>>
>> $\theta \leftarrow \mathrm{step}\left(\theta, \nabla_\theta \widehat{\mathcal{L}}\right)$      ▷ Gradient descent
>> $\gamma \leftarrow \mathrm{step}\left(\gamma, \nabla_\gamma \widehat{\mathcal{L}}\right)$
> **end for**

---

**Variance-exploding (VE) SDE / scaled Brownian motion:**   This SDE is given by a scaled Brownian motion, i.e., $\mu := 0$ and $\sigma$ as defined above. It holds that

$$X_t | X_0 \sim \mathcal{N}\left(X_0, 2\nu^2 \alpha(t)\mathrm{I}\right).$$

For $X_0 \sim \mathcal{N}(m, \Sigma)$, we thus have that

$$X_t \sim \mathcal{N}\left(m, 2\nu^2 \alpha(t)\mathrm{I} + \Sigma\right).$$

A.6   COMPUTATIONAL DETAILS

For convenience, we first outline our method in Algorithm 1. Recall that the methods DIS, PIS, and DDS can be recovered when making particular choices for $v$, $r$, and $p_{\mathrm{prior}}$, see Table 1. We specify the corresponding setting and further computational details in the following.

**General setting:**   Every experiment is executed on a single GPU and, in our PyTorch implementation, we generally follow the settings and hyperparameters of DIS and PIS as presented in Berner et al. (2024), which itself is based on the implementation of Zhang & Chen (2022). In particular, we use the Fourier MLPs of Zhang & Chen (2022), a batch size of $2048$, and the Adam optimizer. To facilitate the comparisons, we use a fixed number of 200 steps for the Euler-Maruyama scheme. A difference to Berner et al. (2024) is that we observed better performance (for all considered methods

and losses) by using an exponentially decaying learning rate starting at 0.005 and decaying every 100 steps to a final learning rate of $10^{-4}$. We use 60000 gradient steps for the experiments with $d \leq 10$ and 120000 gradient steps otherwise to approximately achieve convergence. However, we observed that the differences between the losses are already visible before convergence, see, e.g., Figure 1.

**PIS:** We follow Zhang & Chen (2022) and use a Brownian motion starting at $\delta_0$ for the uncontrolled SDE $X^0$. Furthermore, we also leverage the score of the target density $\nabla \log \rho$ (typically given in closed-form or evaluated via automatic differentiation) for the parametrization of the control $u$, see Zhang & Chen (2022); Berner et al. (2024).

**DIS:** We use the VP-SDE in Song et al. (2020) for the SDE $Y^0$. Specifically, we use $\nu := 1$ and

$$\beta(t) := (1 - t)\beta_{\min} + t\beta_{\max}, \quad t \in [0, 1],$$

with $\beta_{\min} = 0.05$ and $\beta_{\max} = 5$, see Appendix A.5. Moreover, we employ a linear interpolation of $\nabla \log \rho$ and $\nabla \log p_{\text{prior}}$ for the parametrization of the control $u$, see Berner et al. (2024).

**Bridge:** For the general bridge, we consider the loss (7), which corresponds to the setting in Chen et al. (2021a) adapted to unnormalized densities. We use an analogous setting to DIS; however, we additionally employ a Fourier MLP to control the process $Y^v$. Since $Y_T^0$ is already close to $p_{\text{prior}}$ by construction of the VP-SDE, we use a lower initial learning rate of $10^{-4}$ for the control $v$. While these choices already provide better results for the KL divergence, see Table 5, we note that more sophisticated, potentially problem-specific choices might be investigated in future studies. In particular, for the general bridge, we would be free to choose the prior density $p_{\text{prior}}$ as well as the drift function $\mu$ in the SDEs (2) and (3).

**Log-variance loss:** For the log-variance loss, we only change the objective from $\mathcal{L}_{\text{KL}}$ to $\mathcal{L}_{\text{LV}}^w$, where we used the default choice of $w := u$, i.e., $X^w := X^u$. We emphasize that we do not need to differentiate w.r.t. $w$, which results in reduced training times, see Figure 9. In practice, we can thus detach $X^w$ from the computational graph without introducing any bias. This can be achieved by the `detach` and `stop_gradient` operations in PyTorch and TensorFlow, respectively. We leave other choices of $w$ to future research and anticipate that choosing noisy versions of $u$ in the initial phase of training might lead to even better exploration and performance. Furthermore, we use the same hyperparameters for the log-variance loss as for the KL-based loss. As these settings originate form Berner et al. (2024) and have been tuned for the KL-based loss, we suspect that optimizing the hyperparameters for the log-variance loss can lead to further improvements.

**Evaluation:** To evaluate our metrics, we consider $n = 10^5$ samples $(x^{(i)})_{i=1}^n$ and use the ELBO as an approximation to the log-normalizing constant $\log Z$, see Appendix A.6.1. We further compute the (normalized) *effective sample size*

$$\text{ESS} := \frac{\left(\sum_{i=1}^n w^{(i)}\right)^2}{n \sum_{i=1}^n \left(w^{(i)}\right)^2},$$

where $(w^{(i)})_{i=1}^n$ are the importance weights of the samples $(x^{(i)})_{i=1}^n$ in path space. Finally, we estimate the Sinkhorn distance[5] $\mathcal{W}_\gamma^2$ (Cuturi, 2013) and report the error for estimating the average standard deviation across the marginals, i.e.,

$$\text{std} := \frac{1}{d} \sum_{k=1}^d \sqrt{\mathbb{V}[G_k]}, \quad \text{where} \quad G \sim p_{\text{target}}.$$

### A.6.1 COMPUTATION OF LOG-NORMALIZING CONSTANTS

For the computation of the log-normalizing constant $\log Z$ in the general bridge setting, we note that for any $u, v \in \mathcal{U}$ it holds that

$$\mathbb{E}\left[\frac{d\mathbb{P}_{\tilde{Y}^v}}{d\mathbb{P}_{X^u}}(X^u)\right] = 1.$$

---

[5]Our implementation of the Sinkhorn distance is based on `https://github.com/fwilliams/scalable-pytorch-sinkhorn` with the default parameters.

Together with Proposition 2.3, this shows that

$$\log Z = \log \mathbb{E}\left[\exp\left(-\left(R_{f_{u,v,u}^{\text{Bridge}}} + S_{u+v} + B\right)(X^u)\right)\right]. \tag{27}$$

If $u = u^*$ and $v = v^*$, the expression in the expectation is almost surely constant, which implies

$$\log Z = -\left(R_{f_{u^*,v^*,u^*}^{\text{Bridge}}} + S_{u^*+v^*} + B\right)(X^{u^*}). \tag{28}$$

If we only have approximations of $u^*$ and $v^*$, Jensen's inequality shows that the right-hand side in (28) yields a lower bound to $\log Z$. For PIS and DIS, the log-normalizing constants can be computed analogously, see Zhang & Chen (2022); Berner et al. (2024). If not further specified, we use the lower bound as an estimator for $\log Z$ in our experiments.

### A.7 PARTIAL TRAJECTORY OPTIMIZATION

In this section, we present a method that does not need the simulation of entire trajectories but can rely on subtrajectories only. On the one hand, this promises faster computations and, on the other hand, it can be used for exploration strategies since subtrajectories can be started at arbitrary prescribed locations, independent of the control $u$. Crucially, this strategy only works for the log-variance loss and not for the KL-based loss.

Let us recall that the log-variance loss in (8) is defined for any process $X^w$. In particular, in addition to the control $w$, we can also freely choose the initial condition of $X^w$, see the proof of Proposition 2.3. Motivated by Zhang et al. (2023), we can leverage this fact to train the DIS or DDS methods on smaller time-intervals $[t, T]$. Specifically, recall that the log-variance loss for the DIS method in (14) is given by

$$\mathcal{L}_{\text{LV}}^w(u) = \mathbb{V}\left[\left(R_{f_{u,w}^{\text{DIS}}} + S_u + B\right)(X^w)\right], \tag{29}$$

where the optimal control is defined by

$$u^* = \sigma^\top \nabla \log \overleftarrow{p}_{Y^0}, \tag{30}$$

see Section 3.2. Also, recall that

$$B(X^w) = \log \frac{p_{\text{prior}}(X_0^w)}{\rho(X_T^w)}, \tag{31}$$

where $p_{\text{prior}} \approx p_{Y_T^0}$. Now, assuming an approximation $\Phi(\cdot, t) \approx \log p_{Y_t^0}$ for $t \in [0, T]$, we can replace $\log p_{\text{prior}}$ in (31) by $\Phi(\cdot, t)$, and consider the corresponding sub-problems on time intervals $[t, T]$. For the log-variance loss, we can then sample $t \sim \text{Unif}([0, T])$, choose an arbitrary initial condition $X_t^w$, and minimize the loss (29) on the time interval $[t, T]$.

In order to obtain an approximation $\Phi(\cdot, t) \approx \log p_{Y_t^0}$, we could train a separate network, e.g., by using the underlying optimality PDEs in Appendix A.4. However, for the DIS method, this is not needed if we parametrize the control $u$ as

$$u = \sigma^\top \nabla \overleftarrow{\Phi}, \tag{32}$$

where $\Phi$ is a neural network. Based on (30) we can then use $\Phi(\cdot, t)$ as an approximation of $\log p_{Y_t}$ during training. Therefore, we can therefore optimize the loss

$$\mathcal{L}_{\text{LV,sub}}^w(\Phi) := \mathbb{V}\left[\left(R_{f_{\sigma^\top \nabla \overleftarrow{\Phi}, w}^{\text{DIS}}} + S_{\sigma^\top \nabla \overleftarrow{\Phi}} + B_{\Phi,\text{sub}}\right)(X^w)\right],$$

w.r.t. the function $\Phi$. In the above, we can pick $t \sim \text{Unif}([0, T])$, $X_t^w \sim \nu$ with $\nu$ being a suitable probability measure on $\mathbb{R}^d$, and

$$B_{\Phi,\text{sub}}(X^w) := \Phi(X_t^w, t) - \log \rho(X_T^w),$$

where (with slight abuse of notation) the integrals $R$ and $S$ defined in (1) now run from $t$ to $T$.

The subtrajectory training procedure has three potential benefits. First, training may be accelerated since we consider smaller time-intervals $[t, T]$, leading to faster simulation of the SDEs. Second, we can choose $X_t^w$ in a suitable way to prevent mode collapse, e.g., we can sample $X_t^w$ from a

Table 3: We compare the performance of DIS without using the target information $\nabla \log \rho$ in the parametrization. In this case, the performance of the KL-based loss is generally decreasing, as also observed in Zhang & Chen (2022). For the log-variance loss, we can counteract this decrease by relying on sub-trajectories starting at random $t \sim \text{Unif}([0, T])$ and $x \sim \text{Unif}([-a, a]^d)$ (for sufficiently large $a \in (0, \infty)$) in order to facilitate exploration, see Appendix A.7. This allows to obtain competitive results without using any gradient information of the target.

| Problem | Loss | $\Delta \log Z \downarrow$ | $\mathcal{W}_\gamma^2 \downarrow$ | ESS $\uparrow$ | $\Delta \, \text{std} \downarrow$ |
|---------|------|------|------|------|------|
| GMM ($d = 2$) | KL-DIS (Berner et al., 2024) | 2.291 | 3.661 | 0.8089 | 3.566 |
| | LV-DIS-Subtraj. (ours) | **0.059** | **0.020** | **0.8613** | **0.008** |
| DW ($d = 5, m = 5, \delta = 4$) | KL-DIS (Berner et al., 2024) | 3.983 | 5.517 | 0.3430 | 1.795 |
| | LV-DIS-Subtraj. (ours) | **0.394** | **0.121** | **0.4378** | **0.002** |

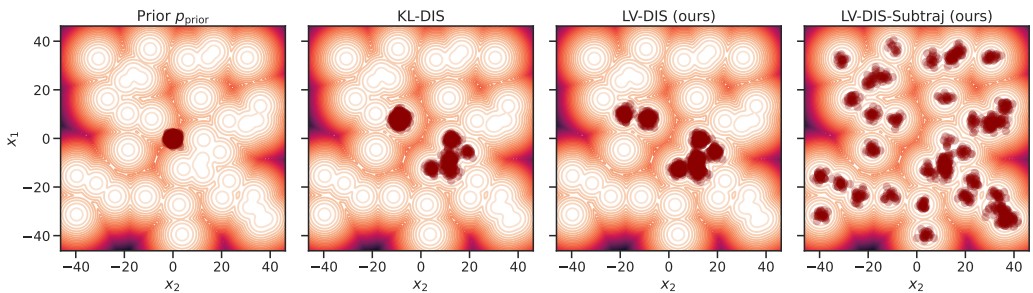

Figure 4: Contour plots of a Gaussian mixture model $p_{\text{target}}$ with $40$ modes analogous to the problem proposed in Midgley et al. (2023). We plot samples of the prior $p_{\text{prior}} = \mathcal{N}(0, \mathrm{I})$ (left) and the DIS method trained with the KL-based loss, the log-variance loss, and partial trajectory optimization (from left to right), see Appendix A.7. For all methods, we use $T = 2$ to guarantee $p_{Y_T^0} \approx p_{\text{prior}}$. Using the setting from Table 3, subtrajectory training can recover all modes without gradient information from the target, whereas other methods suffer from mode collapse—despite making use of $\nabla \log \rho$. Midgley et al. (2023) report mode collapse on this benchmark for several state-of-the-art methods. We remark that LV-DIS (unlike KL-DIS) recovers all modes when slightly increasing the prior variance.

distribution $\nu$ with sufficiently large variance. Third, in (32), we consider a parametrization of $u$ that does not rely on $\nabla \log \rho$, which may not be available or expensive to compute.

In addition to these benefits, we show in Table 3 that subtrajectory training can achieve competitive performance compared to the results in Table 2. On the contrary, we show that the performance of DIS with the KL-based loss gets worse when not using parametrizations that contain the term $\nabla \log \rho$. Note that subtrajectory training cannot be used with the KL-based loss since, for this loss, we need to sample the initial condition according to $X_t^u$.

In Figure 4, we compare the performance on the benchmark proposed in Midgley et al. (2023). We show that partial trajectory optimization can identify all $40$ modes of the Gaussian mixture model, significantly outperforming the DIS method, even when using the log-variance loss and the derivative of the target in the parametrization. We note that Midgley et al. (2023) report mode collapse on this benchmark for other state-of-the-art methods, such as Stochastic Normalizing Flows (Wu et al., 2020), Continual Repeated Annealed Flow Transport Monte Carlo (Arbel et al., 2021), and flows with Resampled Base Distributions (Stimper et al., 2022).

## A.8 FURTHER EXPERIMENTS AND COMPARISONS

In this section, we present further results and ablation studies. In Table 4, we show that the log-variance loss also leads to improvements for smaller batch sizes. This can be motivated by its variance-reducing effect, see Proposition 2.5, Remark A.2, and Figure 5. In Figures 6 and 7, we

Table 4: The same setting as in Table 2 is considered, however, with a smaller batch size of $512$ instead of $2048$. We again observe that the log-variance loss consistently yields better performance. This can be motivated by the inherent variance reduction of its gradient estimators (particularly helpful for smaller batch sizes), see Proposition 2.5, Remark A.2, and Figure 5.

| Problem | Method | Loss | $\Delta \log Z \downarrow$ | $\mathcal{W}_\gamma^2 \downarrow$ | ESS $\uparrow$ | $\Delta \text{std} \downarrow$ |
|---|---|---|---|---|---|---|
| GMM ($d = 2$) | PIS | KL (Zhang & Chen, 2022) | 2.201 | 2.708 | 0.0002 | 3.576 |
| | | LV (ours) | **2.200** | **2.629** | 0.0002 | **3.564** |
| | DIS | KL (Berner et al., 2024) | 1.725 | 0.088 | 0.0045 | 2.711 |
| | | LV (ours) | **0.063** | **0.020** | **0.8517** | **0.004** |
| DW ($d = 5, m = 5, \delta = 4$) | PIS | KL (Zhang & Chen, 2022) | 3.693 | 4.949 | 0.0001 | 1.793 |
| | | LV (ours) | **0.285** | **0.124** | **0.5957** | **0.008** |
| | DIS | KL (Berner et al., 2024) | 4.047 | 5.068 | 0.0015 | 1.797 |
| | | LV (ours) | **0.447** | **0.121** | **0.3917** | **0.002** |

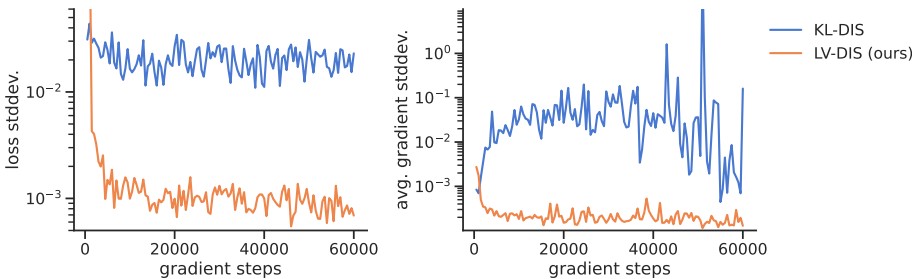

Figure 5: We compare the standard deviations of the loss and (average) gradient estimators using either the KL-based loss or the log-variance loss. Each standard deviation is computed over $40$ simulations of the loss without updating the parameters. We show results for the DIS method on the $5$-dimensional DW target. As predicted by our theory, the log-variance loss exhibits significantly smaller standard deviations for both the loss and its gradient.

show that the log-variance loss can counteract mode collapse in both moderate as well as very high dimensions. Moreover, we present the results for the general bridge approach in Table 5, and we consider a problem from Wu et al. (2020) in Figure 8. In Figure 9, we present boxplots to show that our results from Table 2 are robust w.r.t. different seeds.

Finally, in Table 6, we show that our methods are competitive to other state-of-the-art sampling baselines. However, we want to emphasize that the focus of our work is not to extensively compare against MCMC methods or normalizing flows. Our goal is to show that recently developed methods, such as PIS, DDS, and DIS, can be unified under a common framework, which enables the usage of different divergences. We then propose the log-variance divergence, which makes diffusion-based samplers even more competitive and mitigates potential downsides compared to other methods. The fact that there are general trade-offs between the considered diffusion-based samplers and variants of MCMC has already been discussed and numerically analyzed in the papers introducing the respective methods, see Zhang & Chen (2022); Vargas et al. (2023a).

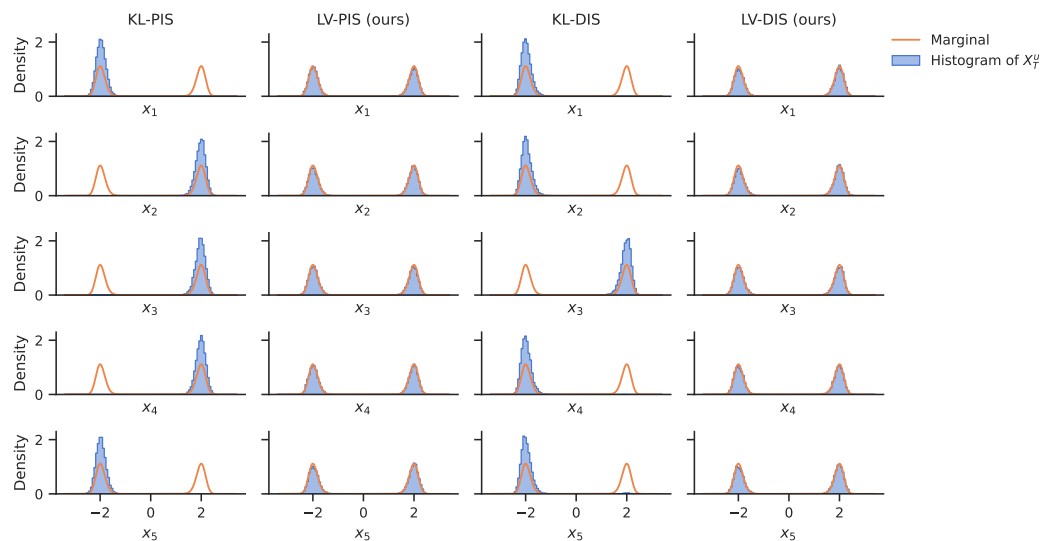

Figure 6: Marginals of samples from PIS and DIS (left and right) for the DW problem with $d = 5$, $m = 5$, and $\delta = 4$. The mode coverage of the log-variance loss is superior to the KL-based loss for all marginals (from top to bottom).

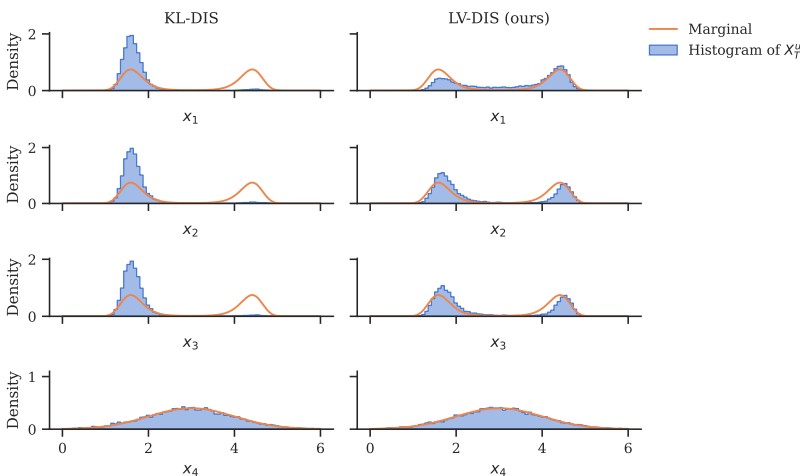

Figure 7: Marginals of the first four coordinates of samples from DIS for a high-dimensional shifted double well problem in dimension $d = 1000$ with $m = 3$ and $\delta = 2$ (see Section 4.1), using the KL or the log-variance loss, respectively. Again one observes the better mode coverage of the log-variance loss as compared to the reverse KL divergence.

Table 5: Metrics of the general bridge approach in Section 2.3 for selected benchmark problems. We observe a clear improvement using the log-variance loss. Moreover, for the KL divergence, we note that the general bridge framework can obtain better results than DIS or PIS, see Table 2. As in Table 2, we report the median over five independent runs. We show errors for estimating the log-normalizing constant $\Delta \log Z$ as well the standard deviations $\Delta \operatorname{std}$ of the marginals. Furthermore, we report the normalized effective sample size ESS and the Sinkhorn distance $\mathcal{W}_\gamma^2$ (Cuturi, 2013). The arrows $\uparrow$ and $\downarrow$ indicate whether we want to maximize or minimize a given metric.

| Problem | Method | Loss | $\Delta \log Z \downarrow$ | $\mathcal{W}_\gamma^2 \downarrow$ | ESS $\uparrow$ | $\Delta \operatorname{std} \downarrow$ |
|---|---|---|---|---|---|---|
| GMM ($d = 2$) | Bridge | KL (Chen et al., 2021a) | 0.328 | 0.393 | 0.0180 | 0.698 |
| | | LV (ours) | **0.084** | **0.020** | **0.9669** | **0.010** |
| DW ($d = 5, m = 5, \delta = 4$) | Bridge | KL (Chen et al., 2021a) | 0.872 | 0.132 | 0.0561 | 0.099 |
| | | LV (ours) | **0.215** | **0.119** | **0.5940** | **0.002** |

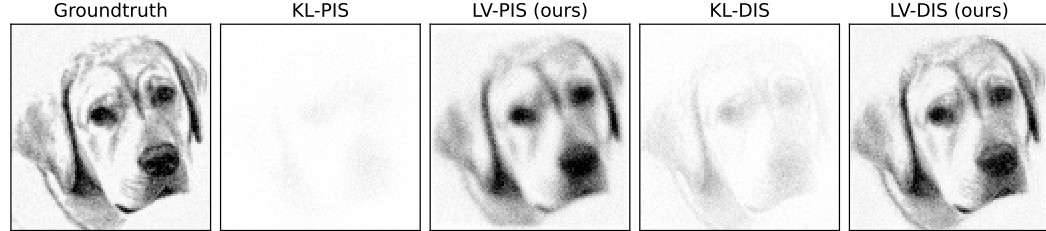

| Groundtruth | KL-PIS | LV-PIS (ours) | KL-DIS | LV-DIS (ours) |

Figure 8: Comparison of samples for the target in Wu et al. (2020). For the KL-based losses, a large fraction of the samples (PIS: 86%, DIS: 67%) lies outside of the domain despite the low density values. On the other hand, the log-variance loss significantly improves performance, yielding competitive performance compared to stochastic normalizing flows presented in Wu et al. (2020).

Table 6: We compare our methods to Continual Repeated Annealed Flow Transport Monte Carlo (CRAFT), see Arbel et al. (2021). We adapt the proposed configurations[6] to use the same batch size and number of iterations as our methods and evaluate all methods using $10^5$ samples. We see that diffusion-based sampling, in combination with the log-variance loss, can provide competitive performance across a range of metrics. We report the median over five independent runs and compare the log-normalizing constant (using the reweighted estimator in (27)), the Sinkhorn distance to ground truth samples, and the error in estimating the average standard deviation.

| Problem | Method | $\Delta \log Z \, (rw) \downarrow$ | $\mathcal{W}_\gamma^2 \downarrow$ | $\Delta \operatorname{std} \downarrow$ |
|---|---|---|---|---|
| GMM ($d = 2$) | CRAFT (Arbel et al., 2021) | 0.012 | 0.020 | 0.019 |
| | LV-PIS (ours) | **0.001** | 0.020 | 0.023 |
| | LV-DIS (ours) | 0.017 | 0.020 | **0.004** |
| Funnel ($d = 10$) | CRAFT (Arbel et al., 2021) | 0.123 | 5.517 | 6.139 |
| | LV-PIS (ours) | 0.097 | 5.593 | 6.852 |
| | LV-DIS (ours) | **0.028** | **5.075** | **5.224** |
| DW ($d = 5, m = 5, \delta = 4$) | CRAFT (Arbel et al., 2021) | 0.001 | **0.118** | **0.000** |
| | LV-PIS (ours) | **0.000** | 0.121 | 0.001 |
| | LV-DIS (ours) | 0.043 | 0.120 | 0.001 |
| DW ($d = 50, m = 5, \delta = 2$) | CRAFT (Arbel et al., 2021) | **0.000** | 6.821 | 0.001 |
| | LV-PIS (ours) | 0.001 | 6.823 | **0.000** |
| | LV-DIS (ours) | 0.422 | 6.855 | 0.009 |

---

[6]The configuration files can be found at `https://github.com/deepmind/annealed_flow_transport/blob/master/configs`.

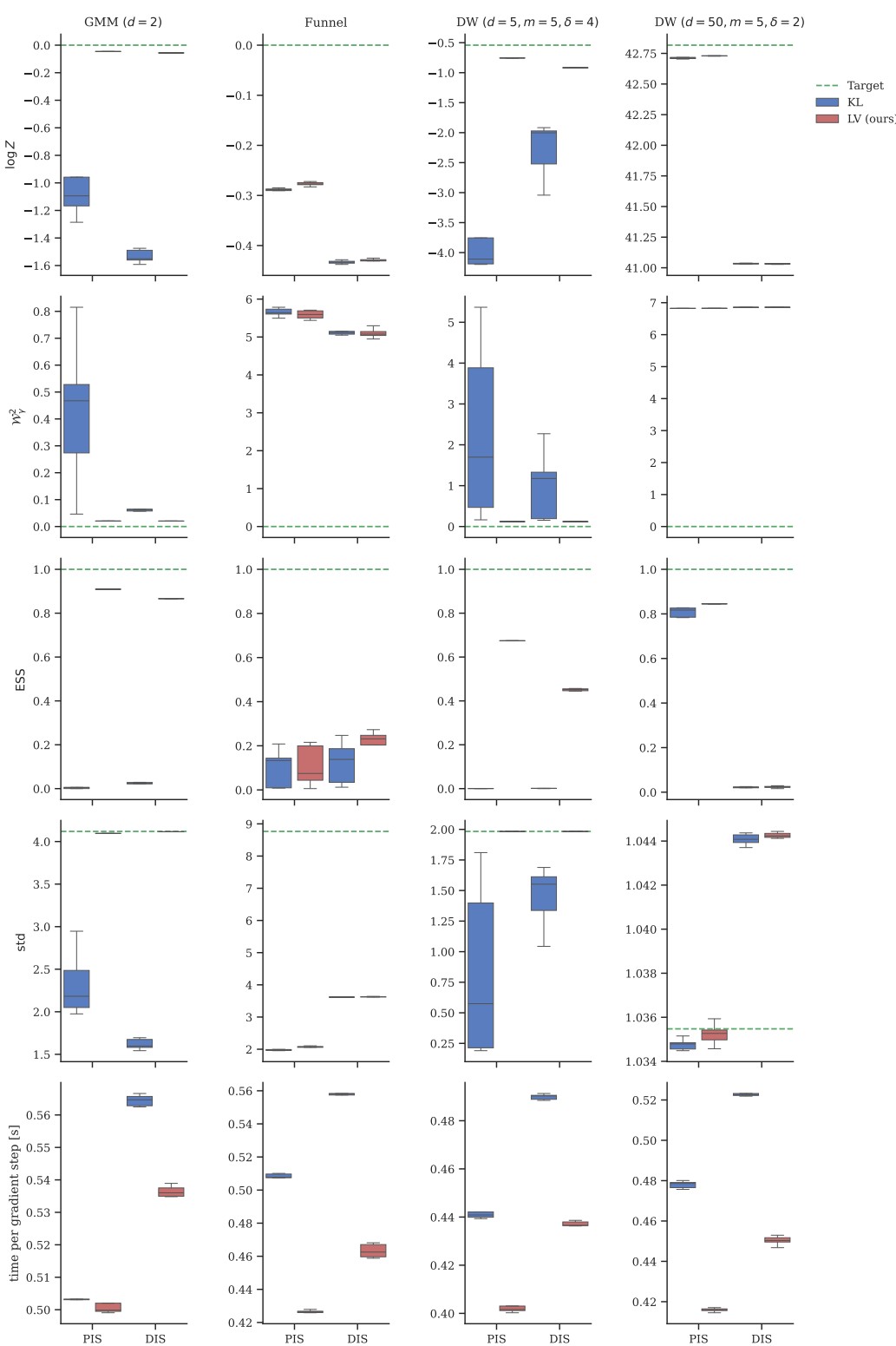

Figure 9: Boxplots for five independent runs for each problem and method (KL-PIS, LV-PIS (ours), KL-DIS, LV-DIS (ours) from left to right in each plot) in the settings of Table 2 and corresponding ground truth or optimal values (dashed lines). It can be seen that the performance improvements of the log-variance loss are robust across different seeds. At the same time, the log-variance loss reduces the average time per gradient step by circumventing differentiation through the SDE solver.

