# OpenReview forum: "Improved sampling via learned diffusions"
_ICLR.cc/2024/Conference — ICLR 2024 poster_

### Official Review · Reviewer_kAh6 · 2023-10-30

**Soundness:** 4 excellent
**Presentation:** 4 excellent
**Contribution:** 3 good
**Rating:** 6
**Confidence:** 4

**Summary:**

The work proposes a unified framework that clearly explains recent sampling algorithms based on generative models. The improvement is led by replacing KL divergence with log variance loss. Experiments show that with new loss, existing generative sampling can achieve better performance.

**Strengths:**

1. Authors present the work and its theoretical analysis clearly, good writing.
2. The usage of the new divergence metric leads to much more robust training/optimization, which is a critical issue for existing learning-based sampling algorithms.

**Weaknesses:**

1. Although the author claims the framework is new. The underlying math theory and contributions have existed long before.
2. To further improve the readability, authors should clearly define the log-variance loss and outline the algorithm pseudocode in the appendix.
3. PIS work shows the network that has information of target distribution gradients can perform much better. Is it also true with new log-variance loss?
4. I am a bit confused discussion of stop gradient, does it lead to a biased gradient estimation? Is it the same trick used in Chen 2021a?
5. Can authors provide more descriptions and analysis on log-variance and its computations? How do authors estimate the mean, which is required to calculate the variance?

**Questions:**

See above

---

> ### Author Response · Authors · 2023-11-20
> **Thank you for you review**
>
> Dear reviewer,
>
> Thank you for your review. We are happy that you rate the soundness and presentation of our paper as excellent and value our contribution of significantly improving sampling performance via our novel framework. Let us address your concerns and questions in the following.
>
> * *The underlying math theory has existed before.* We agree that the theoretical *tools* used in our novel framework are rather classical (e.g., Girsanov theorem, Ito formula, stochastic optimal control, divergences, optimality PDEs and corresponding BSDEs of SBs, etc.). However, to the best of our knowledge, we are the first to leverage these tools to derive the general likelihood for the (continuous-time) path space setting (see Proposition 2.3). This result is one of the key ingredients to deriving our methods. Only through this result, it becomes possible to consider other divergences, such as the log-variance divergence, which has not been applied to the sampling problem before and—according to our numerical results—yields substantial improvements.
>
>     We also think that it is more of an advantage than a disadvantage to explain and develop modern machine learning methods (leading to state-of-the-art results) with known mathematical tools that are well-studied and mature.
>
> * *Outline the algorithm in the appendix.* We added the pseudocode to the appendix in Section A.6. Thank you for making this suggestion, which greatly improves the presentation of our work.
>
> * *Provide more details and analysis on the log-variance loss /  Clearly define the log-variance loss.* Thank you for this hint and suggestion. In the newly added algorithm environment, we explicitly define the empirical version of the log-variance loss.
> In particular, we just compute the empirical variance $\frac{1}{m-1}\sum_{i=1}^m \left(x_i - \frac{1}{m}\sum_{j=1}^m x_j\right)^2$. In PyTorch, this can be done with the command `torch.var`. Note that Bessel's correction is not important since it only rescales the loss. We theoretically introduce the log-variance divergence (and the corresponding log-variance loss) in Definition 2.4 and Equation (8) in the paper.
>
>     We also added further empirical evaluation of the variance-reducing effect of the log-variance loss in Figure 5 in the appendix. This supports our analysis in Proposition 2.5 (robustness at the solution) and Remark A.1 in the appendix (control variate interpretation). Moreover, we refer to [1, 2] for further analysis.
>
>
> * *Networks with target information perform better*. Thank you for this remark. We can confirm the observation made in the PIS paper that it usually helps to incorporate $p_\mathrm{target}$ (or its unnormalized version $\rho$) in the neural network.
>
>      However, different from the KL-based loss, the log-variance loss yields a derivative-free method when omitting the target in the parametrization. More precisely, it can be used for black-box problems where we have only zeroth-order information from $p_\mathrm{target}$. We added a corresponding discussion and experiment in Section A.7 in the appendix. Based on a novel extension of our algorithm leveraging partial trajectory optimization, we can show that we can achieve similar results even in the derivative-free setting.
>
> * *Does "stop gradient" lead to a biased gradient estimation?* Thank you for the question. We now clarify this point in our manuscript in Appendix A.6. Detaching the control $w$ in the computational graph of the log-variance loss does *not* lead to a biased gradient estimation. This comes from the fact that the log-variance divergence actually corresponds to a family of divergences parametrized by the function $w$. Specifically, one can optimize the controls $u$ and $v$ for (almost) arbitrary choices of $w$, all of them yielding unbiased and valid losses. As outlined in the text (and motivated by the control variate interpretation in Remark A.1 in the appendix), we usually choose $w = u$ in practice and detach $u$ from the computational graph (since we do not want to take the derivative w.r.t. to $w$). For the KL-based loss, one *must* set $w=u$, and detaching would lead to a meaningless loss. Since the work in Chen (2021a) is based on the KL divergence, they need to rely on a costly, alternating training scheme when detaching the control.
>
> ---
>
> [1] Nikolas Nüsken and Lorenz Richter. *Solving high-dimensional Hamilton–Jacobi–Bellman PDEs using neural networks: perspectives from the theory of controlled diffusions and measures on path space*. Partial Differential Equations and Applications, 2(4):1–48, 2021.
>
> [2] Lorenz Richter, Ayman Boustati, Nikolas Nusken, Francisco Ruiz, and Omer Deniz Akyildiz. *VarGrad: low-variance gradient estimator for variational inference*. Advances in Neural Information Processing Systems, 33:13481–13492, 2020.

---

### Official Review · Reviewer_29Lh · 2023-10-30

**Soundness:** 3 good
**Presentation:** 2 fair
**Contribution:** 3 good
**Rating:** 6
**Confidence:** 3

**Summary:**

This paper proposes a framework for learning to sample from unnormalized target densities. Specific cases of this framework include several recently proposed algorithms (Schrodinger Bridge, Time-Reversed Diffusion Sampler, Path Integral Sampler, Denoising Diffusion Sampler). The framework involves a general divergence between path measures of diffusion processes. This encompasses the most common case of the reverse KL divergence, but other divergences can also be used. The paper introduces another divergence named the log-variance divergence, which is shown empirically to outperform the KL by avoiding mode collapse.

**Strengths:**

The paper proposes a theoretical framework that encompasses several recently proposed methods for sampling via diffusions. This is a very interesting contribution, in particular from a theoretical point of view, since it allows a unified conceptualization of these methods. The proposed divergence seems to outperform the KL divergence, making it also an interesting contribution to the community.

**Weaknesses:**

[Update on Nov 22: authors have answered in details the various points raised below.]

I found the paper quite dense, although I acknowledge that this is partly due to the nature of the contribution which is to propose a unifying framework. In particular, Section 3 assumes precise knowledge of the various samplers recently proposed in the literature. While it is not a problem in itself, I would suggest in case of lack of space to move part of the derivations of this Section to the Appendix.

**More importantly, I am not convinced by the explanations on the advantages of the proposed log-variance divergence in Section 2.3**, although I agree that the experiments show that it outperforms the KL divergence in the presented settings. This is an important issue in my opinion, especially since these advantages are part of the narrative of the paper (see the introduction and the conclusion).

First, regarding the choice of $w$, it is stated that this choice allows a balance between exploitation and exploration. I do not understand what this precisely means. I believe this statement needs to be backed by either a theoretical analysis or a numerical experiment (or be removed). It is also stated without justification that choosing $w$ different from $u$ might be beneficial to avoid mode collapse. At the same time, in the experiments, $w$ is chosen equal to $u$ (according to Appendix A.4), and yet mode collapse is already avoided. Furthermore, I did not see where it is specified in Section 3 how $w$ should be chosen for the connection with the various approaches to hold.

Regarding the computational benefit, the proposed method still requires to compute gradients with respect to $u$ and $v$ even though gradients wrt $w$ are not required. It is not clear for me why this entails a significant improvement in computational efficiency. I think the naive implementation with stop_gradient still requires backpropagating through the SDE, although I agree that the computational graph is lighter. Experimentally, the time improvement shown in Figure 6 is marginal. Nevertheless, I believe that a more efficient ad hoc implementation that requires only a forward pass through the SDE is perhaps possible. However, this is not explained in the paper. Furthermore, unless I am mistaken, simulating the SDE is still required in any case. This should be mentioned to avoid inducing the reader into thinking that it is not required (like in score matching).

Finally, Proposition 2.5 is not uninteresting, but it is difficult without further explanations or experiments to understand if it plays a role in the improvement observed in practice in the paper.

All in all, I think that either more precise explanations must be given on these various points, or the claims regarding the proposed log-variance divergence must be adapted.

**Minor remarks that do not influence the rating**
+ Proposition 2.3: I think $\rho$ is undefined.
+ The title of the paper is not very informative. I think it would be interesting to consider a title more specifically linked to the contributions of the paper.

**Questions:**

Do the authors think the proposed methodology could be adapted to the case of generative modeling?

One could imagine that the variance of the Monte-Carlo estimator of the variance is high, which could explain why a large batch size (2048 in the experiments) is required. Did the authors try comparing their method with the KL divergence for smaller batch sizes? Do the authors still observe an improvement wrt the KL divergence?

---

> ### Author Response · Authors · 2023-11-20
> **Thank you for your review (1/2)**
>
> Dear reviewer,
>
> We thank you very much for your detailed review and are happy that you value our contribution as being very interesting, allowing for a unified conceptualization of sampling based on diffusions. We also appreciate that you consider our log-variance loss valuable for the ICLR community. Let us respond to your concerns and questions in the sequel.
>
> **Dense presentation**
>
> We agree that our presentation is rather dense, which is indeed also caused by the page limit. We tried to incorporate proper storytelling and moved multiple technical details to the appendix. Section 2 focuses on our main contribution (the variational path space perspective). Section 3 then draws connections to existing methods (Schrödinger bridge, DIS, PIS, DDS). Additionally, we added a high-level explanation of the sampling methods in Section A.3 and pseudocode in Algorithm 1 in the appendix (see the PDF). Moreover, Table 1 helps to understand the technical connections between the different methods. Note that we wrote Section 3 to be self-contained, i.e., so that one can, in principle, understand the precise connections even without knowing the various other methods. This is why we need to rely on some formulas.
>
> If you have more suggestions, we would be very happy to improve our presentation further.
>
> **Clarifications regarding the log-variance divergence**
>
> Thanks for bringing up this topic. We also think that the advantages of the log-variance loss are part of the paper's narrative—therefore, we were keen on making them clearer and are grateful for any further suggestions. In particular, we added more discussion after Definition 2.4, an ablation with a smaller batch size in Table 4, an empirical validation of the variance-reducing effect of the log-variance loss in Figure 5, and an extension of our method based on partial trajectory optimization (actively promoting exploration) in Appendix A.7 (see the PDF). Let us answer your specific questions in more detail in the following:
>
> *Further explanations of the role of Proposition 2.5.*
>
> This is a great suggestion! We added Figure 5, which compares the variance of the loss and its average gradient for the KL and LV divergence. As predicted by our theory, we observe a substantial variance reduction when converging to the optimum for the log-variance loss ("sticking-the-landing") and a rather high, non-vanishing variance for the KL-based loss.
>
> To answer your question, the "variance of the Monte-Carlo estimator of the variance" (for the log-variance loss) is thus actually *lower* than the variance of the Monte-Carlo estimator of the mean (for the KL-based loss). This motivates why the improvements of the log-variance should also hold for lower batch sizes. As suggested by you, we added an ablation for a lower batch size in Table 4, which verifies this claim.
>
>
> *Backpropagation through SDE*.
>
> You mentioned that you think "a more efficient ad hoc implementation that requires only a forward pass through the SDE is possible." The log-variance loss indeed only requires a forward pass (and no backward pass) through the SDE. When we simulate the process $X^w$, we detach $w$ from the computational graph such that no gradient of the stochastic process $X^w$ has to be computed. The SDE must still be simulated to evaluate respective quantities and, e.g., compute the derivatives w.r.t. $u$ and $v$. However, unlike the KL case, no gradients of the stochastic process have to be computed.
> In particular, we can use any black-box SDE solver and do not need to track the derivatives "through the SDE solver" for automatic differentiation.
>
> To give an example, the losses usually contain the term $\log \rho(X_T^w)$ to incorporate the target information—for the KL-based losses, one needs to compute derivatives of this quantity, whereas, for the log-variance loss, one does not. This *derivative-free setting* is crucial for problems where the target is only available as a black box.
>
> We consider a corresponding experiment in Appendix A.7 and added further explanations below Definition 2.4.

---

> ### Author Response · Authors · 2023-11-20
> **Thank you for you review (2/2)**
>
> *Role of the forward process $X^w$*.
>
> Thank you for bringing up this point.
>
> The connections in Section 3 hold for any forward process $X^w$, but the exact algorithms from previous works (i.e., DIS, PIS, DDS) rely on the KL divergence, which requires $w = u$ by design. In fact, it is a contribution of our work that one can now also approach the methods DIS, PIS, and DDS with the log-variance loss and an arbitrary forward control $X^w$.
>
> We note that the improved performance of the log-variance loss does not only come from making choices of $w$ different from $u$. As you correctly observe, also the standard choice $w=u$ yields better results (and counteracts mode collapse). First, recall that the log-variance loss yields a provably zero variance of its gradients for any $w$ *at the optimum*.
> However, for the choice $w=u$, the derivative w.r.t. $u$ can generally be interpreted as a control variate version of the derivative of the KL divergence (see Remark A.1 in the appendix).
> We added a remark on this important fact below Proposition 2.5.
>
> We also mention that, in the case of the log-variance loss, one has not only freedom in picking the control $w$ but also the initial condition of the process $X^w$. Motivated by recent work (Zhang et al., 2023), we use this fact to devise an extension of our method, which uses trajectories starting at random points on the space-time domain, thus actively promoting exploration (in particular, including low probability regions of $X^u$). We achieve this by optimizing over subtrajectories of $X^w$ on $[t,T]$ for uniformly chosen $t$ and uniformly chosen starting point $X^w_t$, where we use the partially learned density as new prior. We refer to the new section in Appendix A.7 for further details.
>
> By construction, this approach reduces simulation time. Moreover, we rely only on zero-th-order information of the target and can still (slightly) improve our best results for the DIS method. In comparison, the KL-loss is not even amendable to such a derivative-free setting, and we show in Table 3 that omitting the target in the parametrization negatively affects performance (as also observed by Zhang & Chen (2022)).
>
>
> **Minor remarks**
>
> *Proposition 2.3: I think $\rho$ is undefined.* We added the definition of $\rho$ below Proposition 2.3. We think that the unnormalized density $\rho$ was actually defined in the first equation in the introduction. But we agree that the reader should again be reminded of its definition.
>
> *The title of the paper is not very informative.* Thank you for the suggestion. Upon acceptance, we will discuss the possibility of changing the title with the AC.
>
> ---
>
> We are grateful for your constructive suggestions, which significantly improved our manuscript.
> We hope that we have addressed all your concerns. If so, we would be very happy if you consider updating your score. If not, please let us know what else we could improve. Thank you again!

---

> > ### Comment · Reviewer_29Lh · 2023-11-22
> >
> > Thanks to the authors for their thorough answers. I have increased my score acccordingly.

---

> > > ### Author Response · Authors · 2023-11-23
> > > **Thank you for your update**
> > >
> > > We are glad that we could answer your questions. Thank you again for the helpful and constructive review, which significantly improved our paper.

---

### Official Review · Reviewer_i9dA · 2023-10-31

**Soundness:** 3 good
**Presentation:** 3 good
**Contribution:** 2 fair
**Rating:** 6
**Confidence:** 2

**Summary:**

The authors propose a unifying framework for various existing diffusion-based samplers. A log-variance based (rather than KL-based) loss is proposed. The proposed work is evaluated against competing methods on the problem of sampling without data --- i.e., with respect to an unnormalized density function.

**Strengths:**

- There is an extensive theoretical discussion of the provided method and its analytical properties.
- The authors provide a useful guide to organize recent works on diffusion-based approaches to density-based sampling.
- The work appears to be mathematically sophisticated and relatively rigorous.

**Weaknesses:**

- Limited validation. The generalizations proposed are compared on rather simplistic examples. Though the authors appear to target scenarios where data is not available, data-based modeling is clearly an important application of a diffusion-based sampler. Is there a reason the models are not compared to other non-diffusion methods, e.g., MCMC, normalizing flow, autoregressive, or GAN models?

- Unclear abstract: First, the abstract appears to claim that the authors "identify [diffusion models] as special cases of the Schrödinger bridge problem". This feels far too bold of a claim to me, as many papers have connected diffusion models to Schrodinger Bridges (and the authors are clearly aware of this). After all, nearly all modern generative diffusion models sample from a target density. Second, the authors focus entirely on the problem of sampling without data -- i.e., w.r.t. a density function. This is in stark contrast to what a general ICLR reader may consider as sampling in diffusion models --- i.e., sampling from a model trained on data. Although it appears that the proposed framework can perhaps be formulated to include this case, the goal of the article is clearly the former and not the latter. In fact, there do not appear to be any meaningful experiments on the latter case. This should be clarified in the abstract, as this fact was not at all clear except for a sentence in the third paragraph of the introduction.

**Questions:**

Please see weaknesses.

---

> ### Author Response · Authors · 2023-11-20
> **Thank you for your review**
>
> Dear reviewer,
>
> We thank you for your review and are happy that you value our theoretical as well as practical contributions. Let us respond to your comments and questions in the sequel.
>
> **Limited validation**
>
> * *Rather simplistic examples are considered*. We agree that, building upon our principled framework for diffusion-based samplers, future work should tackle even more complicated problems. Our empirical validation shows a clear advantage of our proposed methods and, compared to previous diffusion-based sampling methods (e.g., DIS, DDS, PIS), we already made significant improvements. See also our new ablation studies and extensions to partial trajectory optimization in the appendix.
>
>     We add that, from a sampling point of view, our examples are already quite challenging. Sampling from multimodal (unnormalized) densities with isolated modes is known to be hard, even in moderate dimensions. In fact, the Funnel and DW problems, as well as complex GMMs, are considered challenging benchmark problems for recently developed methods. This is underlined by the performance of state-of-the-art methods on our considered benchmarks (see Table 6 in the appendix).
>
>     Let us also refer to Figure 8 in the appendix, which considers a 1000-dimensional DW example, showing that the effect of the log-variance loss does translate even to very high dimensions. We note that the DW problems resemble typical and relevant problems in molecular dynamics, where potentials contain multiple energy barriers and resulting local minima correspond to meaningful configurations of a molecule.
>
> * *Data-based modeling is also important*. We agree that the setting of having data samples available, but no density, is also an interesting problem. Although some advancements of one setting can be translated to the respective other, there are crucial differences. In particular, note that a direct computation of divergences between path space measures seems to be not possible in data-based modeling since the analytical expression of the target density (which, in this setting, is usually not available) appears in the related Radon-Nikodym derivative (see Proposition 2.3). However, we agree that future work can now explore the benefit of additionally using our objectives for the (arguably rare) scenario where both data and an unnormalized density are available.
>
> * *Comparison to other sampling methods*. Note that, additionally to other diffusion-based samplers, we have compared our results to a number of alternative sampling methods in Table 6 in the appendix. We observe that we can outperform state-of-the-art methods based on MCMC, normalizing flows, and SMC. To the best of our knowledge, GANs are not applicable in the scenario where one only has access to an unnormalized density, since the discriminator relies on data samples. Standard autoregressive models seem to be used for density estimation tasks, and typical models for continuous state spaces can not be used for sampling. However, we are very open to comparing to further baselines suggested by you.
>
> **Unclear abstract**
>
> We thank the reviewer for pointing out this aspect and agree that we can improve the clarity of our abstract. We adjusted the abstract accordingly to emphasize that we are interested in *sampling from unnormalized densities* rather than from a distribution that is (only) specified by data samples (as in the classical diffusion-based *generative modeling* case, where one does not have access to densities).
>
> We also agree that we are definitely not the first to connect diffusion-based sampling to Schrödinger bridges. We have mentioned several related references in the text and also the fact that our work was inspired by [1]—however, with our path space measure perspective, one can see that the problem formulation is non-unique, which is not discussed in [1]. By this argument, we admit that the abstract was not exact in the sense that we are considering a *generalized Schrödinger bridge* that only demands some bridge between prior and target, not necessarily minimizing $\operatorname{KL}(\mathbb{P}\_{X^u}| \mathbb{P}\_{X^0})$—thank you for the good catch.
>
> We have adapted the abstract accordingly (see the updated PDF). Let us know if you think it is more suitable now or if you would like to see further changes.
>
> ---
>
> Thanks again for improving the overall readability and clarity with your suggestions!
> If we addressed all your concerns, we would be happy if you consider potentially updating your score. Otherwise, we would be happy if you let us know of any further improvements.
>
> ---
>
> [1] Chen, T., Liu, G. H., \& Theodorou, E. A. (2021). Likelihood training of Schrödinger bridge using forward-backward SDEs theory. ICLR 2022.

---

### Official Review · Reviewer_y7Cc · 2023-11-03

**Soundness:** 4 excellent
**Presentation:** 4 excellent
**Contribution:** 4 excellent
**Rating:** 8
**Confidence:** 4

**Summary:**

The paper tackles sampling problems, such as sampling from unnormalized densities (i.e., target distributions), by using stochastic differential equations (SDEs).

To do that, the paper first proposes a novel framework to unify previous SDEs-based sampling methods: (1) variational formulation of the Schrödinger bridge (SB) problem, (2) path integral sampler (PIS), (3) time-reversed diffusion sampler (DIS), and (4) denoising diffusion sampler (DDS). In the framework, the authors show that we can obtain the analytic form of the Radon-Nikodym (RN) derivative of one SDE wrt another SDE, evaluated at any path generated by the third SDE, as long as all SDEs have the same noise coefficient term (while time direction can vary). Then, the authors show that we can define proper divergences between two SDEs using the RN-derivative (by taking expectation with the RN-derivative). Furthermore, the authors show that the previous SDE-based sampling problems can be defined as the minimization of the reverse KL divergence (or its variant) wrt a reference measure; more specifically, the choice of the reference measure and forward will determine the corresponding sampling problems.

Second, based on the framework, the authors propose to use the log-variance divergence instead of the reverse KL divergence thanks to several benefits of the log-variance against the reverse KL; for example, the log-variance divergence uses a reference measure and thus does not require to differentiate through the SDE solver. The paper shows all four variants based on the log-variance divergence corresponding to SB, PIS, DIS, and DDS.

Finally, the paper demonstrates the performance of the proposed methods against the previous works by using some benchmark datasets.

**Strengths:**

In my understanding, the paper's contributions are clear, and I also consider that the results are essential for several reasons:

1. The paper proposes a novel framework that provides better understanding of previous literature on sampling problems using SDEs.
2. The paper well motivates the log-variance divergence-based methods so that readers can understand how each step contributes to the merits of the proposed method.
3. I found that the paper has a well-organized structure that makes it clear to understand the proposed method. Significantly, the paper shares sufficient information to understand the new divergence’s behavior, which will help readers understand relevant backgrounds and the proposed method.

In general, the paper's contributions wrt the novelty are clear, and the proposed methods are well-defined. In addition, I found that the paper has a well-organized structure that makes it clear to understand the proposed methods.

**Weaknesses:**

In general, I find that the paper is well-written. However, descriptions of some derivations can be improved for clarification. For example, to introduce the derivation of Equation (10), the paper uses “defined in (2) with $u$ replaced by $r \in \mathcal{U}$”. I find this description a little confusing, as Equation 10 assumes three SDEs. Clarifying such descriptions would be helpful to potential readers who are not familiar with SDEs and relevant backgrounds.

**Questions:**

N/A

---

> ### Author Response · Authors · 2023-11-20
> **Thank you for your review**
>
> Dear reviewer,
>
> We thank you for your extensive review and your very good and precise summary of our work. We are happy that you value both our contributions and our presentation. We particularly appreciate that you agree with the significance of the log-variance divergence and that you conclude that our novel (and more general) framework will be helpful in the further understanding and development of diffusion-based sampling methods.
>
> Thank you for commenting on potential minor notational issues. In particular, in the example you pointed out, we now refer to the appendix (due to space restrictions), where we added further clarifications.
> More general, we improved the descriptions in the main part and included more explanations and additional material (extensions of our method and ablation studies) in the appendix, see the PDF.
>
> We thank you again for your comment which helped us to improve our manuscript! Feel free to hint at further notational improvements, which we would gladly like to incorporate.

---

### Author Response · Authors · 2023-11-23
**Thank you for the reviews**

We thank all reviewers again for their very constructive and helpful reviews. We are glad they all believe our contribution is valuable to the ICLR community. We also appreciate the excellent ratings on our work's soundness, presentation, and contribution.

**Improvements**

The reviews helped us to improve both our contribution and presentation further. In the updated PDF, we added a detailed algorithm environment (Algorithm 1), elaborated more on the strengths of the log-variance loss (Section 2.2), and added further empirical evidence. In particular, we provided ablations for other batch sizes (Table 4) and visualized the variance reduction during training (Figure 5). We also devised a new method that only relies on subtrajectories (Appendix A.7), further speeding up training and improving our results even in the derivative-free setting (Table 3). In the present revision, we added one more experiment (Figure 4), showcasing the performance improvements in the benchmark proposed by Midgley et al. (2022).


**Summary**

Finally, let us summarize the contributions of our work which have been reflected in the reviews:

*Unifying framework.*
Our work provides a unifying framework from the perspective of time-reversals, generalizing previous diffusion-based sampling methods (DIS, PIS, DDS, Schrödinger bridges). This framework allows us to formulate the sampling problem as a variational problem over divergences of suitable path space measures.
In addition to novel theoretical insights, this opens the door to using arbitrary divergences on the path space, circumventing known problems of the reverse KL divergence.


*Numerical improvements via log-variance divergence.*
Specifically, we suggest to use the *log-variance divergence*, providing the following advantages (over the KL divergence):
  - It can be interpreted as a variance-reduced version of the KL loss and has the "sticking-the-landing" property.
  - One can freely choose the reference process, allowing one to consider partial trajectory optimization and balance exploitation and exploration.
  - It can be used with any (black-box) SDE solver and does not require tracking the computation of the forward process for automatic differentiation. In particular, no derivatives of the target are required.

---
We believe we incorporated all of the reviewers' recommendations in our rebuttal and are happy to answer any remaining questions.

---

### Meta-Review · Area_Chair_AXzn · 2023-12-03

**Metareview:**

The reviewers agree that this is a good submission and that it should be accepted.

There were some points raised by some of the reviewers, and that have been adressed by the authors. I encourage the authors to take into account the valuable comments from the reviewers in their camera-ready version.

**Justification For Why Not Higher Score:**

Based on other papers that I have seen at ICLR, both submissions this year and previous years, I think that this paper should be accepted, but that it does not raise to the level of a spotlight/oral, mostly in terms of novelty.

**Justification For Why Not Lower Score:**

I have very much enjoyed this submission, and found it to be very interesting. The reviewers all agree that it should be accepted.

---

### Decision · Program_Chairs · 2024-01-16

Accept (poster)